REPORT

# Neutrophil serine proteases degrade endothelial cortactin and promote extravasation

Idaira M. Guerrero-Fonseca[1], Karina B. Hernández-Almaraz[1], Iliana I. León-Vega[1], Régis Joulia[2,3], Armando Montoya-García[1], Hilda Vargas-Robles[1], Theresia E.B. Stradal[4], Klemens Rottner[4,5], Reyna Oregon[6], Eduardo Vadillo[6], Jennifer L. Johnson[7], William B. Kiosses[7], Sergio D. Catz[7], Sussan Nourshargh[2], and Michael Schnoor[1]

**The adhesive interactions of neutrophils with postcapillary venules during inflammation have been well studied. However, how neutrophils trigger molecular changes in endothelial cells (EC) during their extravasation requires further exploration. The endothelial actin-binding protein cortactin regulates endothelial contacts and neutrophil–endothelial interactions, but the associated mechanisms remain elusive. Hypothesizing that endothelial cortactin dynamics change during inflammation, using super-resolution confocal microscopy of inflamed mouse cremasteric venules and HUVEC, we report that neutrophil interaction with EC induces reduction in EC cortactin levels. This response was specifically mediated by neutrophil serine proteases, including cathepsin G, that were detected inside EC. The observed cortactin degradation was abolished after inhibition of serine proteases or blockade of neutrophil exocytosis. Finally, the endogenous serine protease inhibitor α1-antitrypsin suppressed cortactin degradation *in vivo* and reduced neutrophil adhesion and extravasation. Collectively, our data unveil a new mechanism by which neutrophils manipulate proteins inside EC to facilitate their extravasation.**

## Introduction

Neutrophil migration from venules into inflamed tissues is an essential part of host defense against injury and infections (Nourshargh and Alon, 2014; Schnoor et al., 2021). During extravasation, the venular endothelium actively guides neutrophils by upregulating adhesion molecules, producing and presenting chemokines, and forming adhesive membrane structures (Schnoor, 2015; van Steen et al., 2020). Remodeling of the actin cytoskeleton drives the endothelial morphological changes required to allow neutrophil transmigration. Leukocyte adhesion to the apical endothelium via β2-integrin–ICAM-1 interactions results in ICAM-1 clustering and recruitment of actin-binding proteins (ABPs) such as cortactin, α-actinin-4, filamin-B, ezrin, radixin, and moesin that stabilize adhesive interactions (Schaefer et al., 2014; Barreiro et al., 2002). These ABPs support the actin remodeling required for the formation of actin-rich protrusions that engulf and guide adherent leukocytes (Barreiro et al., 2002; Carman and Springer, 2004; Carman et al., 2003; van Buul et al., 2007; Phillipson et al., 2008; van Rijssel et al., 2012; Schnoor et al., 2011). Clustered ICAM-1 also triggers signaling pathways that induce the formation of transmigratory pores at

intercellular contacts. These structures are supported by contractile-actin filaments that enable the paracellular passage of neutrophils while maintaining endothelial integrity and limiting excessive plasma leakage (Heemskerk et al., 2016; Saito et al., 2002; van Steen et al., 2020). The opening of cell contacts and passage of neutrophils requires the disassembly of the VE–cadherin–catenin complex that is connected to the actin cytoskeleton and stabilized by different ABPs, including cortactin. However, the exact mechanisms that control endothelial actin cytoskeleton rearrangements during neutrophil transendothelial migration remain poorly understood.

Cortactin is an ABP that localizes to many actin-rich structures, including lamellipodia and cell–cell junctions, where its main function is to stabilize actin filaments (Schnoor et al., 2018). Cortactin deficiency results in junction destabilization due to reduced Rap1 activity and increased ROCK-mediated MLC phosphorylation, leading to contractile stress fiber formation and hyperpermeability (Schnoor et al., 2011; García Ponce et al., 2016). On the other hand, total cortactin deficiency reduced neutrophil transendothelial migration due to defective RhoG-mediated ICAM-1 clustering, leading to unstable adhesive

[1]Department of Molecular Biomedicine, Cinvestav-IPN, Mexico-City, Mexico; [2]William Harvey Research Institute, Faculty of Medicine and Dentistry, Queen Mary University of London, London, UK; [3]National Heart and Lung Institute, Imperial College London, London, UK; [4]Department of Cell Biology, Helmholtz Centre for Infection Research, Braunschweig, Germany; [5]Division of Molecular Cell Biology, Institute for Cell- and Neurobiology, Technical University Braunschweig, Braunschweig, Germany; [6]Oncology Research Unit, Oncology Hospital, National Medical Center Siglo XXI, IMSS, Mexico-City, Mexico; [7]Department of Molecular and Cellular Biology, The Scripps Research Institute, La Jolla, CA, USA.

Correspondence to Michael Schnoor: mschnoor@cinvestav.mx.

interactions of neutrophils with endothelial ICAM-1 (Schnoor et al., 2011). Collectively, these findings, together with data from early studies (Garcia et al., 1988; Garcia et al., 1998; Baluk et al., 1998; Hurley, 1963), demonstrated that endothelial permeability and neutrophil transendothelial migration are independently controlled processes. Hypothesizing that cortactin functions are spatiotemporally regulated during neutrophil transendothelial migration, we analyzed cortactin dynamics in endothelial cells (EC) during neutrophil extravasation. Here, we show that cortactin in EC is proteolytically degraded by neutrophil serine proteases (NSP) over time during neutrophil–endothelial interactions. We provide evidence for the ability of neutrophils to induce molecular changes inside EC and that this cellular cross talk facilitates breaching of the vascular endothelium by neutrophils during inflammation.

## Results and discussion

### Endothelial cortactin is reduced in postcapillary venules during inflammation

Cortactin is known to localize at EC junctions and to cluster around adherent neutrophils during inflammation (Schnoor et al., 2011). However, the spatiotemporal dynamics of endothelial cortactin during inflammation *in vivo* are unknown. Here, we investigated the expression and localization of cortactin in mouse cremaster postcapillary venules (PCVs) by confocal microscopy. Under basal conditions, cortactin was highly expressed in EC of PCVs (Fig. 1 A). Considerable expression was also detected in extravascular cells (Fig. 1 A, white asterisks). Using Imaris software to extract the endothelial cortactin signal, we detected cortactin presence in the cytoplasm and at endothelial junctions in partial colocalization with CD31 (Fig. 1 B). Quantification of the mean fluorescence intensity (MFI) of the total EC cortactin signal vs the cortactin signal at EC junctions revealed enrichment of cortactin at EC junctions (Fig. 1 C). Cortactin stabilizes actin filaments at its subcellular localizations (Schnoor et al., 2018), and cortactin deficiency in EC results in disrupted barrier function and increased endothelial permeability (Schnoor et al., 2011; García Ponce et al., 2016). Thus, the specific accumulation of cortactin at endothelial junctions in PCVs likely contributes to junction stability.

To investigate if cortactin expression and localization changes during inflammation, cremaster muscles were stimulated with the pro-inflammatory cytokine TNFα. Surprisingly, the cortactin signal diminished 2 h after TNFα stimulation (Fig. 1, D and E). While the reduction of cortactin signal was apparent 30 min after TNFα stimulation, the signal was almost completely lost after 2 h without recovery after 4 h (Fig. S1, A and B). Similar results were obtained when we analyzed cremaster muscles stimulated by a range of other pro-inflammatory mediators. Specifically, we observed significantly reduced levels of cortactin in PCVs when inflammation was induced by TNFα (56.5 ± 3.5% reduction), IL1-β (68.7 ± 4.8%), or CXCL1 (63.6 ± 11.9%) for 2 h and leukotriene-B4 (LTB4) (71.4 ± 3.8%) for 1 h, which was accompanied by robust neutrophil recruitment (Fig. 1, D and E). Interestingly, stimulation of cremaster muscles with histamine for 1 h that does not induce neutrophil recruitment in PCVs did not lead to a statistically significant impact on EC cortactin levels (Fig. 1, D and E).

Spearman's correlation analysis showed a significant association between neutrophil transmigration and reduced levels of EC cortactin (Fig. 1 F; and Fig. S1, C–F).

Additionally, imaging of inflamed cremaster muscles using a 10×-objective showed that reduced EC cortactin level is aligned with venular regions supporting neutrophil adhesion and extravasation (Fig. 2 A, white squares). No change in cortactin levels was noted in the same vessels at sites devoid of neutrophil interaction (Fig. 2 A, orange squares). Furthermore, we did not detect any changes in cortactin expression in neighboring vessels in close apposition to extravasated, interstitial neutrophils (Fig. 2 B, green squares). Collectively, we show that EC cortactin is selectively reduced in inflamed PCVs at sites of neutrophil extravasation. These findings are aligned with the concept that most neutrophils transmigrate paracellularly in PCVs (Woodfin et al., 2011; Vadillo et al., 2019) that requires a temporal loosening of EC contacts, a response likely dependent on reduction of cortactin.

### Reduced EC cortactin protein level is neutrophil dependent

To directly investigate the functional role of neutrophils in the regulation of EC cortactin, mice were subjected to neutrophil depletion by i.v. injection of anti–Gr-1 antibody using a protocol that led to 92.8 ± 3% depletion of the blood neutrophil population (Ly6G+ SSChigh) (Fig. S1, G and H). The administration of an isotype control antibody (IgG2b) or anti–Gr-1 antibody (α–Gr-1) to control mice subjected to intrascrotal (i.s.) PBS injection did not affect the expression of EC cortactin in PCVs (Fig. 2, C and D). In agreement with our previous data, in mice treated with IgG2b, local injection of TNFα led to a significant reduction in EC cortactin expression, but, importantly, this response was abrogated in neutrophil-depleted mice (Fig. 2, C and D). Together, these data indicate that reduction of EC cortactin during inflammation is neutrophil dependent and likely occurs during close interaction of neutrophils with EC during extravasation.

### Neutrophils, not mononuclear leukocytes, degrade endothelial cortactin *in vitro*

Next, confluent monolayers of human umbilical vein EC (HUVEC) were used as an *in vitro* approach to further investigate the mechanism of reduced cortactin levels. Initial experiments showed that treatment of HUVEC with TNFα alone over a range of time points that involved periods of high ICAM-1 expression did not cause any change in cortactin expression (Fig. S2 A). By contrast, when human peripheral blood neutrophils were cocultured with HUVEC for 15 min, we noted a significant reduction in EC cortactin levels as analyzed by flow cytometry (Fig. 2 E, EC gate: cortactin+/CD45− in Fig. S2 B). The loss of cortactin was induced by neutrophils in both unstimulated EC (60.61 ± 4.85% of degradation) and EC pre-treated for 18 h with TNFα (78.41 ± 4.98% of degradation) (Fig. 2 E). The disrupted expression of cortactin showed a tendency (albeit not statistically significant) of being stronger in TNFα-treated ECs. As observed *in vivo*, we noted a significant correlation between reduced EC cortactin signal and the presence of adherent neutrophils per EC (Fig. 2 F). To investigate whether this response is neutrophil

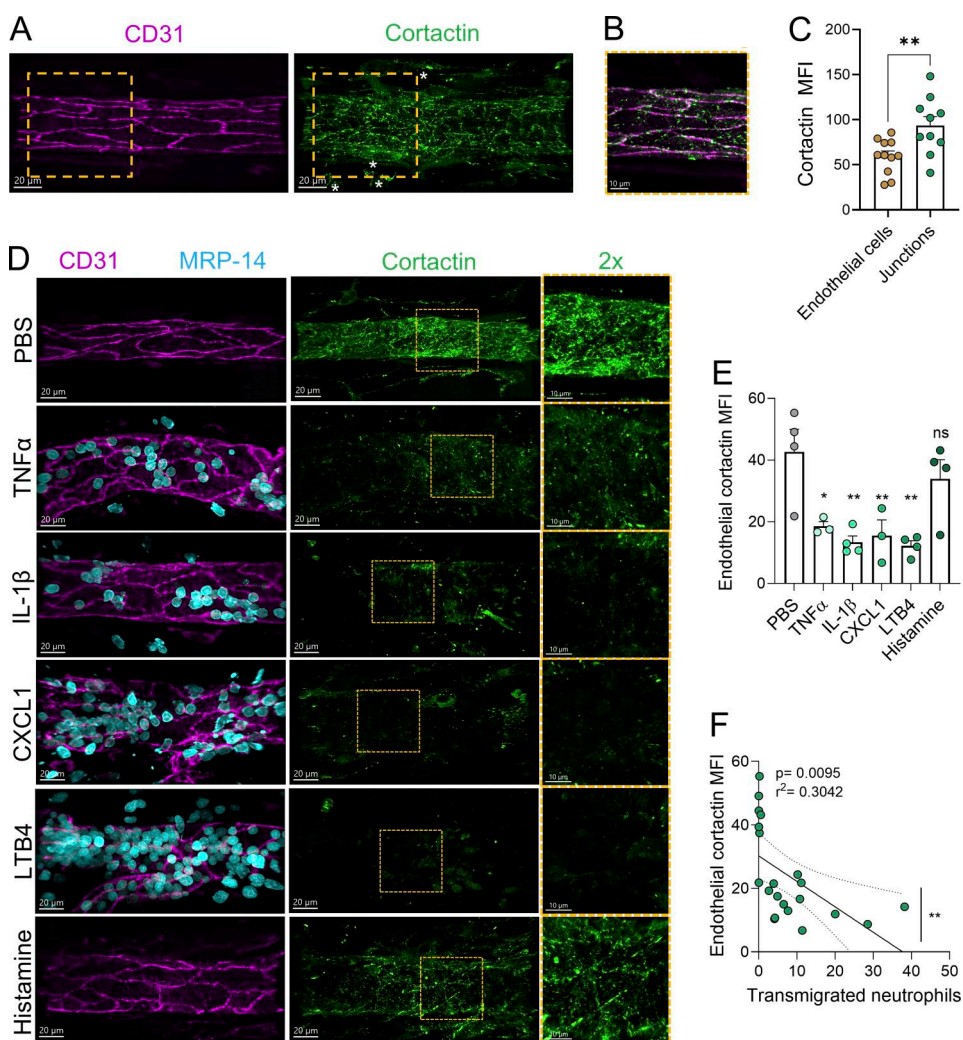

Figure 1. **Cortactin is degraded in PCVs in response to inflammatory stimuli that induce neutrophil recruitment. (A)** Representative 3D confocal images (Leica SP8) of cremasteric PCVs stained for CD31 (magenta) and cortactin (green) (Scale bars = 20 μm). Asterisks indicate cortactin signal in extravascular cells. **(B)** Merge of endothelial cortactin and CD31 from 1.5× zoomed-in orange areas (scale bar = 10 μm). **(C)** Quantification of cortactin MFI in whole venular EC and specifically at EC junctions using Imaris (*n* = 3 mice per group). **(D–F)** Cremaster muscle inflammation was induced by i.s. injection of TNFα, IL1-β, CXCL1 for 2 h, or histamine and LTB4 for 1 h. PBS was injected as control. **(D)** Representative 3D confocal images (Leica SP8) of inflamed cremasteric PCVs stained for CD31 (magenta), MRP-14 (cyan, neutrophils), and cortactin (green); scale bars = 20 μm. The right panels show cortactin in 2× zoomed-in orange areas; scale bars = 10 μm. **(E)** Quantification of cortactin MFI in venular EC using Imaris software (*n* = 3–4 mice/group). **(F)** Correlation analysis of the number of extravasated neutrophils and endothelial cortactin MFI. Each dot represents the average of each mouse of all experimental groups in A. Data are represented as means ± SEM; *P < 0.05; **P < 0.01; ns, not significant.

specific, human peripheral blood mononuclear cells (MNC) were co-cultured with TNFα-stimulated, confluent HUVEC monolayers for 30 min and 1 h. However, MNC co-cultured with EC did not cause loss of cortactin (Fig. 2 G). Together, these findings further support a key role for neutrophils in disruption of EC cortactin levels.

## NSP specifically degrade endothelial cortactin

As calpain and the proteasome have been reported to degrade endothelial cortactin (Stamatovic et al., 2015; Perrin et al., 2006), we speculated that the loss of EC cortactin *in vitro* and *in vivo* is due to enzymatic degradation. However, inhibitors of calpains, the proteasome, or the lysosome in HUVEC before adding neutrophils did not prevent loss of cortactin (Fig. S2, C–F),

suggesting that these pathways are not activated by neutrophil–EC interactions to degrade cortactin.

Intrigued by previous reports of transfer of enzymatically active proteases from neutrophils to EC (Jerke et al., 2015), we hypothesized that cortactin is degraded by neutrophil proteases. To investigate this possibility, putative cleavage sites and corresponding protease families were predicted using Protease Specificity Prediction Server (PROSPER) (Fig. S3 A). This strategy identified 61 cleavage sites for serine proteases and 21 cleavage sites for metalloproteases as the most abundant putative cleavage sites in the cortactin protein sequence. To analyze whether endothelial cortactin is a target for NSP, neutrophils were treated with different pharmacological serine protease inhibitors for 30 min before co-culture with HUVEC. Cortactin

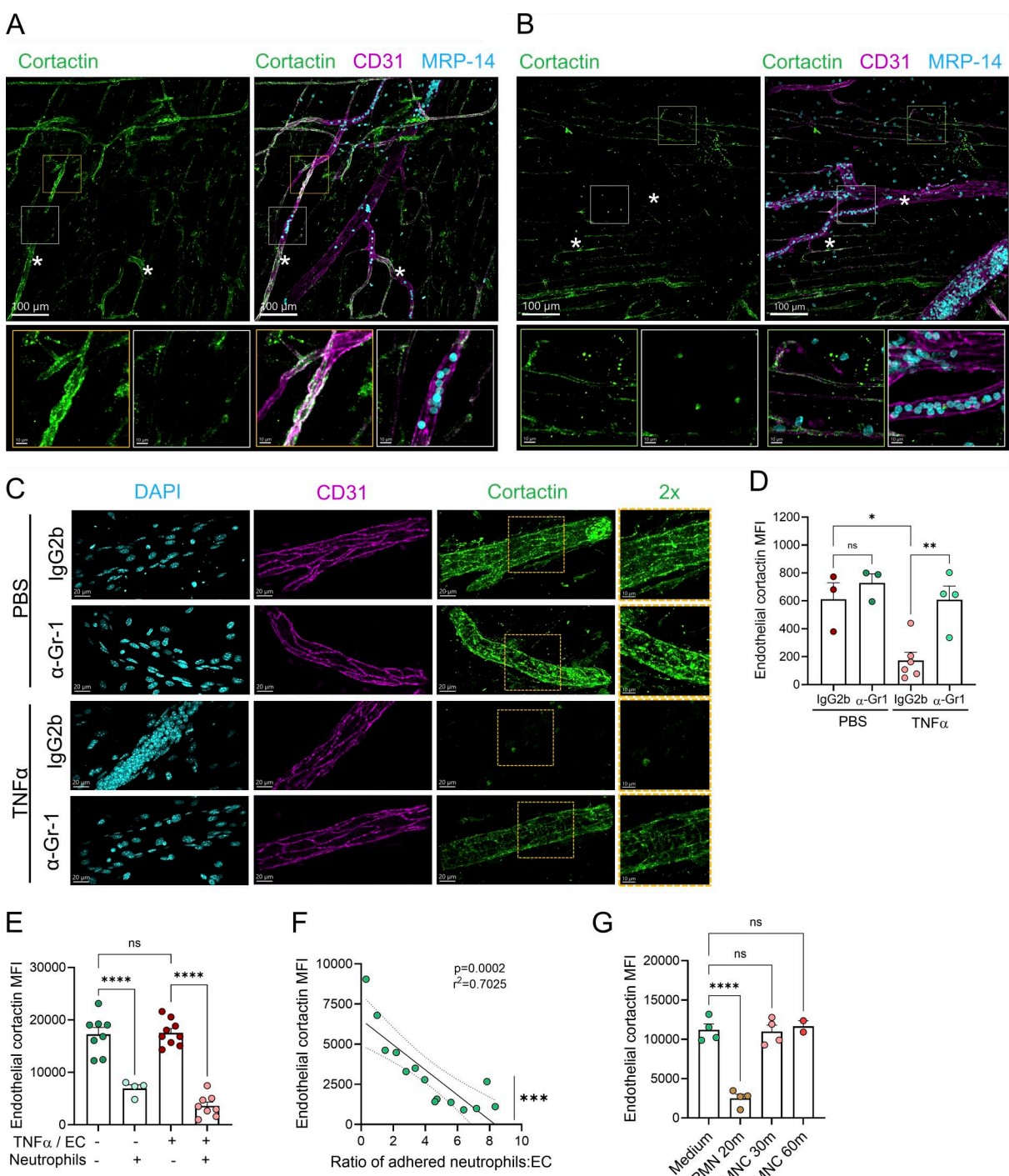

Figure 2. **Neutrophils induce endothelial cortactin degradation *in vivo* and *in vitro*. (A and B)** Representative 3D-confocal images (Nikon A1) of cremaster muscles stimulated with TNFα for 4 h and stained for CD31 (magenta), MRP-14 (cyan), and cortactin (green); scale bars = 100 μm. PCVs (white asterisks) are indicated. Bottom panels show zoomed-in orange, white, and green areas; scale bars = 10 μm. **(A)** Cortactin degradation in PCV is detected when intravascular neutrophils are observed (white squares). PCV devoid of neutrophil interaction (orange squares) do not show cortactin degradation. **(B)** Extravascular neutrophils do not induce cortactin degradation in other vessels (green squares). **(C and D)** Representative Z-stack confocal images (Nikon A1) of cremasteric PCV stimulated with TNFα or PBS for 2 h from neutrophil-depleted mice (α–Gr-1) and mice injected with isotype control antibody (rat IgG2b); scale bars = 20 μm. Right panels show cortactin in 2× zoomed-in orange areas; scale bars = 10 μm. **(D)** Quantification of cortactin MFI in PCV from the images in C (*n* = 3–6 mice/group). **(E–G)** Flow cytometric quantification of cortactin MFI in HUVEC co-cultures. Human peripheral blood neutrophils (1 × 10⁶/ml) or MNC (1 × 10⁶/ml) were co-incubated with confluent untreated or TNFα-treated HUVEC monolayers for 20 min. **(E)** Cortactin MFI in the EC population with or without neutrophil co-incubation (*n* = 4–9 independent experiments). **(F)** Spearman's correlation analysis of EC cortactin MFI and the ratio of adhered neutrophils per EC was performed using the flow cytometry data in E. **(G)** Cortactin MFI in EC after co-culture with neutrophils (PMN, 20 min) or MNC (30 min and 1 h) (*n* = 2–4 independent experiments). Data are represented as means ± SEMs; ∗P < 0.05; ∗∗P < 0.01; ∗∗∗P < 0.001; ∗∗∗∗P < 0.0001; ns, not significant.

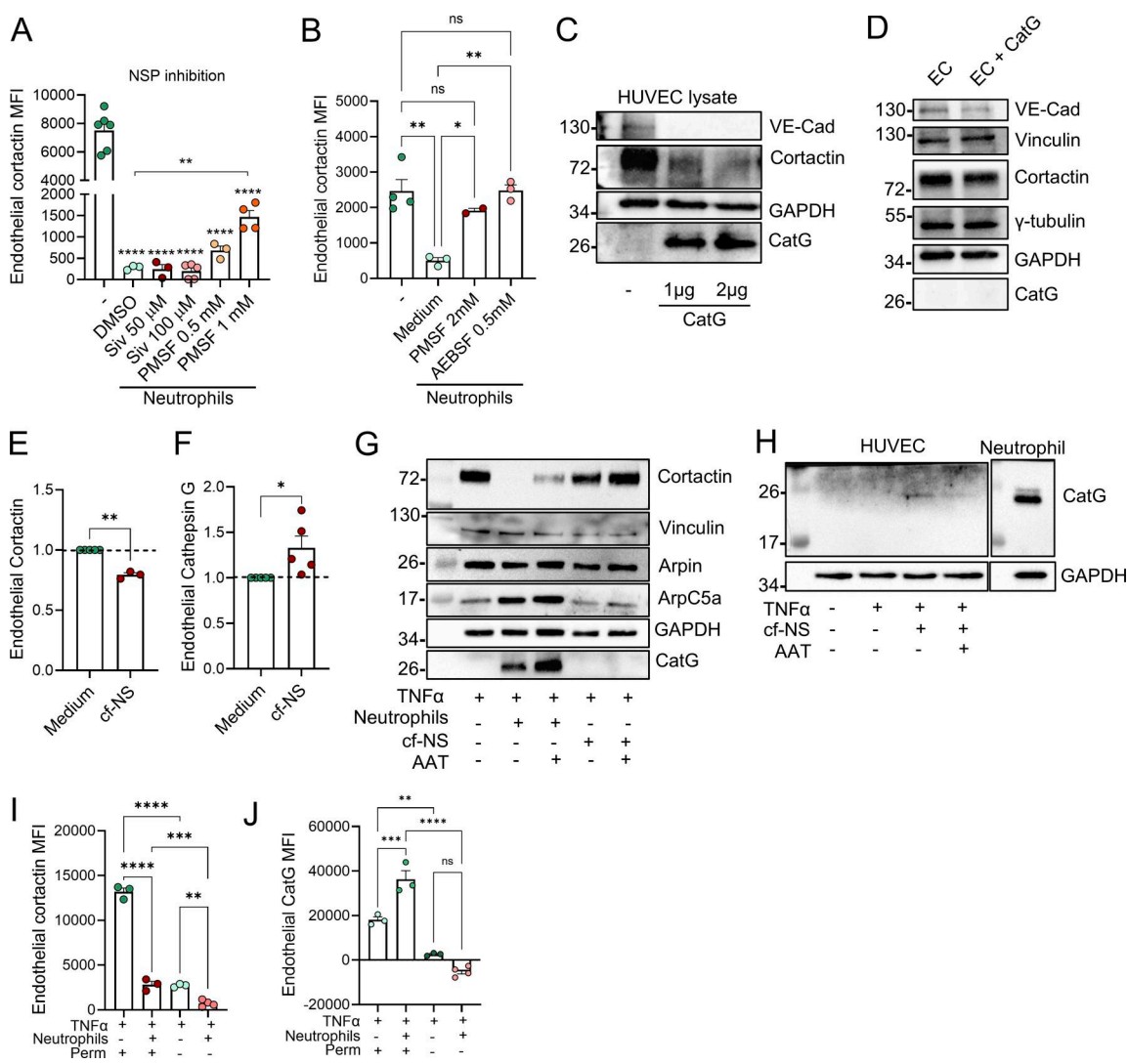

Figure 3. **Endothelial cortactin is degraded by NSP. (A and B)** Flow cytometric quantification of cortactin MFI in HUVEC with NSP inhibition. Neutrophils (1 × 10⁶/ml) were treated with the NE inhibitor Siv, the general serine protease inhibitors PMSF and AEBSF, and vehicle (DMSO or RPMI medium) for 30 min before co-incubation with activated HUVEC monolayers for 15 min. HUVEC without neutrophils (–) were used as control of basal cortactin levels ($n$ = 3–5 independent experiments). **(C)** Western blot (VE-Cadherin, cortactin, GAPDH, and CatG) from HUVEC lysates incubated with 1 or 2 μg CatG for 20 min on ice ($n$ = 3). **(D)** Western blot (VE-Cadherin, vinculin, cortactin, γ-tubulin, GAPDH, and CatG) from HUVEC monolayers incubated with 10 μg/ml CatG for 1 h before extensive washing and protein extraction ($n$ = 3). **(E and F)** Flow cytometric quantification of endothelial cortactin (E) and CatG (F) levels from TNFα-treated HUVEC monolayers that were incubated with cell-free neutrophil supernatant (cf-NS) or medium alone for 1 h ($n$ = 3–5 independent experiments). **(G)** Western blot of actin-related proteins from HUVEC monolayers after co-culture with neutrophils or cf-NS and with or without AAT treatment (10 μg/ml) ($n$ = 3). **(H)** Overexposed western blot from HUVEC monolayers incubated with cf-NS to detect CatG ($n$ = 3). **(I and J)** Flow cytometric quantification of endothelial cortactin (I) and CatG (J) MFI from HUVEC and neutrophil co-cultures under flow conditions. Flow cytometry was performed under permeabilized (Perm⁺) or non-permeabilized conditions (Perm⁻) ($n$ = 3–5). Data are represented as means ± SEM; *$P$ < 0.05; **$P$ < 0.01; ***$P$ < 0.001; ****$P$ < 0.0001; ns, not significant. Source data are available for this figure: SourceData F3.

degradation was not prevented when neutrophil elastase (NE) was inhibited with the specific inhibitor Sivelestat (Siv, ONO 5046). However, the general NSP inhibitors PMSF and AEBSF significantly reduced the loss of cortactin (Fig. 3, A and B), indicating that EC cortactin degradation is mediated by NSP. The fact that Siv alone cannot prevent the loss of cortactin suggests that cortactin can be degraded by multiple NSP, a conclusion supported by our PROSPER analysis (Fig. S3 A).

In testing whether a free NSP can degrade cortactin, we found that incubation of HUVEC lysates with purified human cathepsin G (CatG) led to degradation of cortactin (Fig. 3 C). VE-

cadherin used here as positive control was also degraded, whereas GAPDH used as negative control was not degraded. By contrast, when incubating intact HUVEC monolayers with CatG, we detected partial degradation of the transmembrane protein VE-cadherin, but we could neither detect the presence of CatG in HUVEC after washing and lysis, nor degradation of cortactin or other intracellular proteins such as vinculin, tubulin, or GAPDH (Fig. 3 D). Collectively, these results suggest that NSP can degrade cortactin, but that free NSP are unable to enter intact HUVEC, raising the question of how NSP can enter EC. Given that neutrophils can exocytose vesicles and granules into their

environment (Catz and McLeish, 2020), we investigated whether secreted products in neutrophil supernatants can also degrade endothelial cortactin. Indeed, incubation of HUVEC with cf-NS from activated neutrophils significantly degraded cortactin (Fig. 3 E) and led to the presence of CatG in the EC gate as detected by flow cytometry (Fig. 3 F). These crucial results demonstrate that (1) the observed cortactin degradation is not an artefact of neutrophil lysis, which is known to cause nonspecific degradation of proteins, and (2) that NSP are released from neutrophils before being taken up by EC.

Moreover, western blots for different ABP showed that cortactin was also degraded after co-culture of HUVEC with cf-NS although to a lower degree compared with neutrophil co-culture (Fig. 3 G). Under the same conditions, other intracellular, actin-related proteins such as vinculin, arpin, and the Arp2/3 complex (ArpC5A) were not degraded (Fig. 3 G), and the endogenous NSP inhibitor α1-antitrypsin (AAT) partially inhibited cortactin degradation (Fig. 3 G). CatG could be detected in HUVEC lysates after co-incubation with both neutrophils (Fig. 3 G) and cf-NS (Fig. 3 H). While more intracellular proteins need to be tested in the future, the current data suggest that secretion of neutrophil products leads to a selective degradation of EC cortactin.

This conclusion is supported by analysis of neutrophil–endothelial interactions under physiological shear stress conditions. Specifically, investigating unstimulated and TNFα-stimulated HUVEC exposed to flow rates of 50 dynes/cm$^2$, we observed strong cortactin expression (Fig. S3 B). Under these conditions, the addition of human neutrophils resulted in robust neutrophil–HUVEC interactions (Fig. S3 C) and triggered loss of cortactin signal (Fig. 3 I) and presence of CatG (Fig. 3 J) in HUVEC, as detected by flow cytometry. Analysis of non-permeabilized cells by flow cytometry showed almost undetectable levels of cortactin (Fig. 3 I) and CatG (Fig. 3 J) in HUVEC, suggesting that the neutrophil-derived enzyme is internalized by HUVEC and not simply exposed on the cell surface (Fig. 3 J). Flow chamber assays using whole blood showed a comparable EC cortactin signal with the levels observed after co-incubation with isolated neutrophils (Fig. S3 D). These results indicate that other blood cells such as MNC or soluble blood components do not significantly contribute to the observed cortactin degradation response (compare with Fig. 2 G).

### Neutrophils deliver NSP inside EC

To degrade the intracellular ABP cortactin, NSP would have to be internalized by EC during neutrophil recruitment as also suggested by the above flow chamber data. As NSP, including CatG, are primarily stored in azurophilic granules, we hypothesized that CatG-containing azurophilic granules are exocytosed from neutrophils near EC during transendothelial migration, leading to CatG uptake by EC and cortactin degradation. To address, human neutrophils were treated with Nexinhib20 (NEI20), a Rab27 inhibitor that selectively inhibits exocytosis (Johnson et al., 2016). Subsequently, neutrophils were stimulated with TNFα and fMLP, cf-NS were transferred to HUVEC monolayers, and cells were analyzed by super-resolution confocal microscopy. This strategy revealed a significant amount of CatG inside HUVEC and reduced cortactin levels only when HUVEC were

incubated with cf-NS derived from stimulated neutrophils (Fig. 4 A). 3D analysis of the confocal z-stacks using Imaris software showed that the CatG signal appeared along the strictly intracellular cortactin signal (Fig. 4 B). Importantly, endothelial CatG presence and loss of cortactin was abrogated when HUVEC were incubated with supernatant of stimulated neutrophils pretreated with NEI20 (Fig. 4, A–D). These results suggest that exocytosis inhibition prevents release of NSP from neutrophils and consequently cortactin degradation in HUVEC (Fig. 4 A, and quantification in Fig. 4, C and D).

To extend our *in vitro* studies to inflamed tissues *in vivo*, we analyzed control and TNFα-stimulated cremaster muscles by super-resolution confocal microscopy. In control tissues (PBS-treated), no CatG could be detected (Fig. 4 E). In TNFα-inflamed cremaster muscles, the CatG signal was strong inside and outside of PCVs (Fig. 4 F). Importantly, by extracting the CatG signal from EC using Imaris, we found that a significant amount of CatG was associated with TNFα-treated endothelium (Fig. 4, E and F, bottom). Rendered isosurfaces of CD31 from super-resolution confocal imaging using Imaris showed that much of the CatG signal was indeed located within the plane defined by the endothelial CD31 signal (Fig. 4, E and F, right, and quantification in Fig. 4 G and Video 1). This analysis provides evidence for the presence of CatG within EC of inflamed cremaster venules.

NSP, including CatG, represent a highly toxic, antimicrobial weapon of neutrophils when they are released into the microenvironment upon infection. While the antimicrobial activity of NSP is essential for host survival, its prolonged, excessive secretion causes tissue damage. The internalization of NSP and processing of intra- and extracellular endothelial substrates has been previously described *in vitro* in HUVEC and the EC line ECV304 (Jerke et al., 2015; Yang et al., 2001; Pendergraft et al., 2004; Preston et al., 2002). In vivo, to date, only EC surface receptors are reported to be targets for NSP (Colom et al., 2015). For example, cleavage of EC JAM-C by NE in murine cremaster muscles upon ischemia/reperfusion injury and LTB4 stimulation caused neutrophil reverse transmigration (Colom et al., 2015), and proteolysis of VE-cadherin by CatG and NE or serum from patients with idiopathic inflammatory myopathies containing NSP led to increased endothelial permeability (Hermant et al., 2003; Carden et al., 1998; Gao et al., 2018). Our data now identify cortactin as an intracellular substrate of NSP *in vivo* and *in vitro* during inflammation. Whether cortactin is targeted by different NSP requires further investigation. It is important to highlight that the specific NE inhibitor Siv failed to diminish the degradation of cortactin and that the general NSP inhibitors PMSF and AEBSF prevented cleavage, suggesting that inhibition of one NSP can be compensated by others. This finding together with the fact that the neutrophil exocytosis inhibitor NEI20 also significantly inhibited cortactin degradation argues for the therapeutic importance of inhibiting neutrophil exocytosis over inhibition of individual NSP. This view is supported by data from previous work demonstrating protective effects of NEI20 in a mouse model of systemic inflammation (Johnson et al., 2016).

The mechanism through which EC internalize NSP requires exploration. However, our findings that neutrophil exocytosis inhibition blocks cortactin degradation and that the naked CatG is

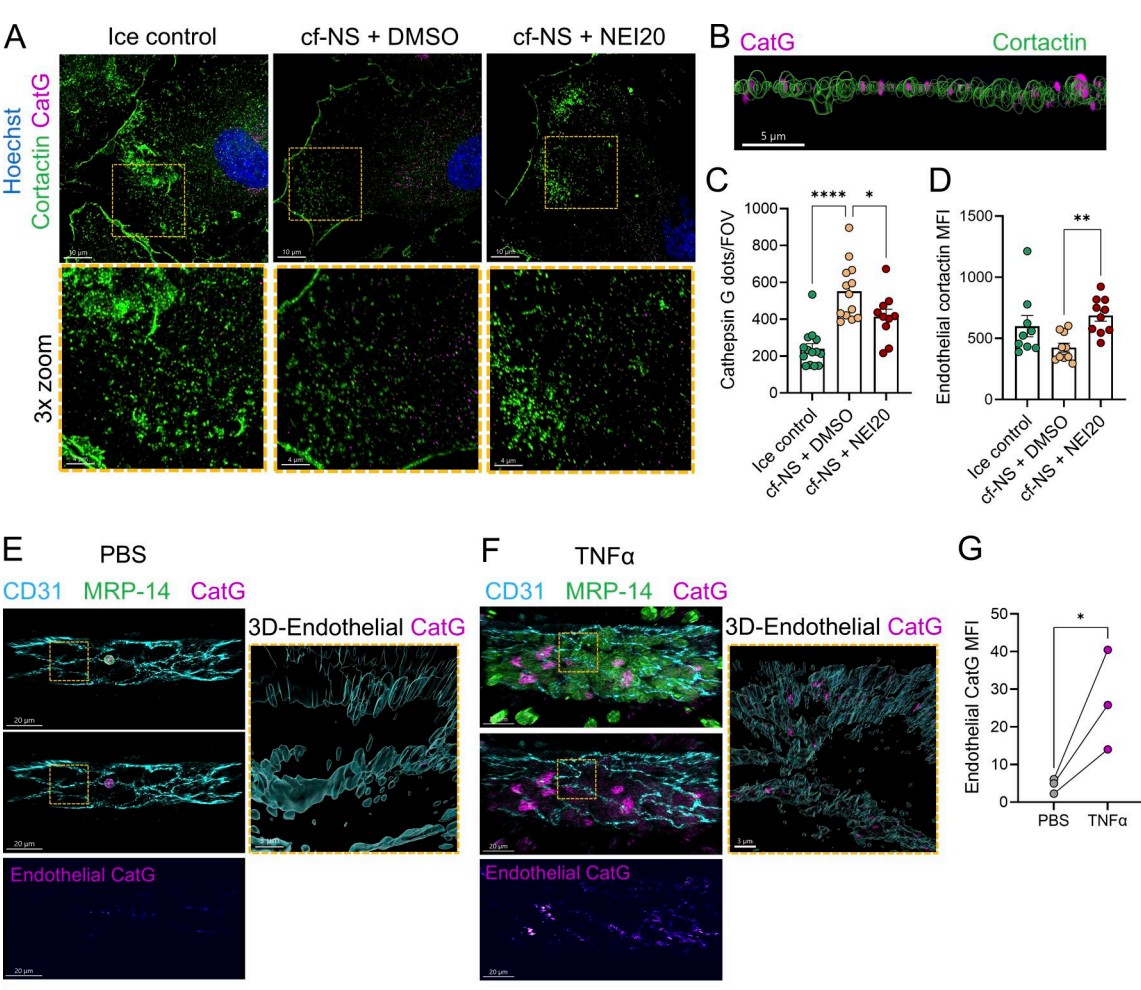

Figure 4. **Neutrophils deliver CatG into EC *in vivo* and *in vitro*. (A)** Representative 3D super-resolution confocal images (Zeiss Airyscan) of HUVEC incubated with cell-free neutrophil supernatant (cf-NS) of non-stimulated (ice control) or stimulated (50 ng/ml TNFα + 1 μM fMLP) neutrophils pre-treated with DMSO or NEI20; scale bar = 10 μm; magnification = 4 μm. **(B)** 3D-rendered isosurface of the endothelial monolayer from the 0.35 μm z-stack images using Imaris software showing that the CatG (magenta) appears along the strictly intracellular cortactin signal (green); scale bar = 5 μm. **(C)** Quantitative analysis of CatG inside HUVEC per field of view (FOV). **(D)** Quantification of the cortactin MFI of the images in A using Imaris software (*n* = 3 independent experiments). **(E and F)** Representative 3D confocal super-resolution images (Leica SP8) of PBS (E) or TNFα-treated (F) cremasteric PCVs (scale bars = 20 μm; magnification = 3 μm). The middle panels have the green channel (MRP-14) removed for clarity. The bottom panels show CatG signals extracted from venules using the CD31 signal. Orange zoomed-in panels show rendered 3D isosurface of endothelial CD31 (cyan) with internalized CatG signal (magenta). **(G)** Quantification of endothelial CatG MFI in PCV; *n* = 3 mice/group. Data are represented as means ± SEM; *P < 0.05; **P < 0.01; ****P < 0.0001.

not internalized by EC and does not induce cortactin degradation argue for a mechanism involving uptake of NSP-containing neutrophil extracellular vesicles via (receptor-mediated) endocytosis (Kolonics et al., 2020; Hong, 2018). Interestingly, intravital microscopy (IVM) studies revealed the release of EV-like elongated neutrophil-derived structures (ENDS) formed by neutrophils rolling on P- and E-selectin (Marki et al., 2021). ENDS contained proteinase 3 and CatG. While the fate or functional relevance of ENDS remains elusive, they could contribute to the transfer of NSP into EC to degrade cortactin.

### NSP inhibition *in vivo* prevents cortactin degradation and reduces neutrophil extravasation

Given that pharmacological inhibition of NSP and exocytosis abrogated the degradation of cortactin, we tested the notion that

inhibiting EC cortactin degradation *in vivo* may suppress neutrophil extravasation. For this purpose, we treated mice with the specific clinical-grade NSP inhibitor AAT, which is the most abundant endogenous NSP inhibitor in the blood with important anti-inflammatory functions (Korkmaz et al., 2010). Patients with AAT deficiency are more susceptible to developing inflammatory diseases, including chronic obstructive pulmonary disease, cirrhosis, and colitis (Greene et al., 2016). AAT has been used in humans to increase the plasma levels of AAT in patients with AAT deficiency and related emphysema (AAT augmentation therapy) and in many experimental murine models of inflammation (Mohanka et al., 2012; Greene et al., 2016; Koulmanda et al., 2008). While cortactin degradation was observed in TNFα-inflamed cremasteric PCVs, this response was significantly inhibited in mice pre-treated with i.p.

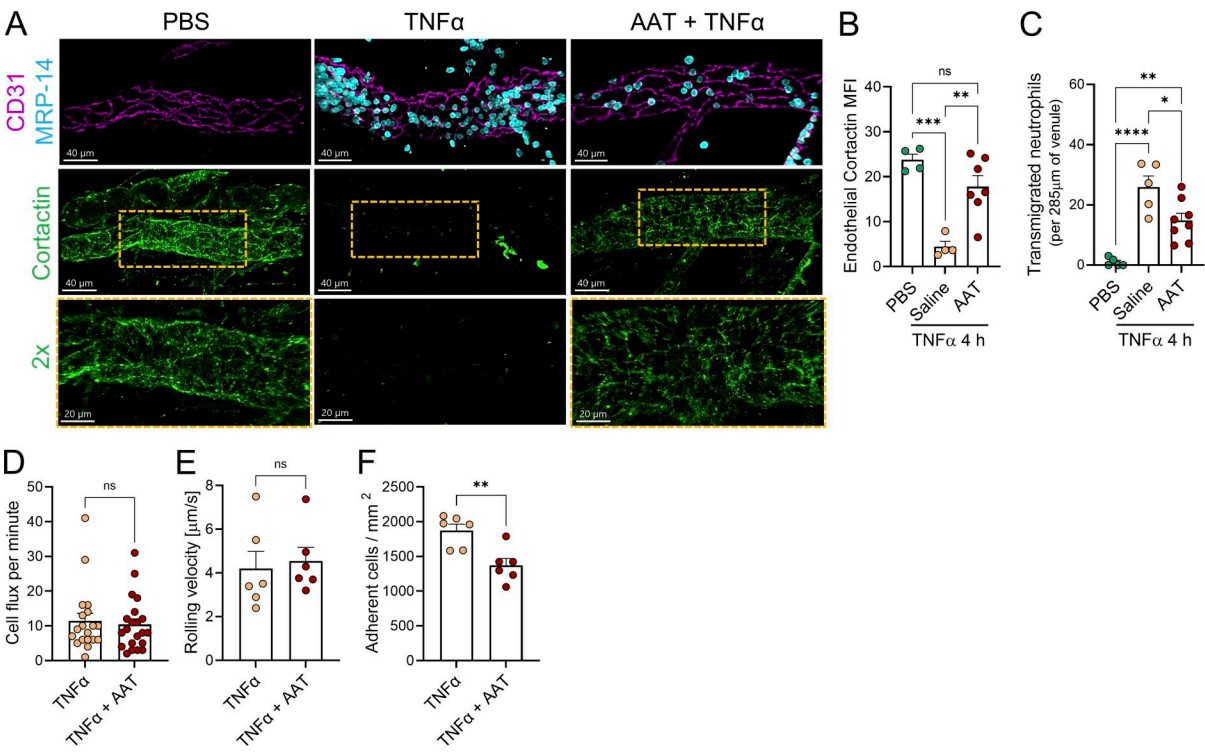

Figure 5. **NSP inhibition by AAT reduces endothelial cortactin degradation and neutrophil extravasation. (A)** Representative z-stack confocal images (Leica SP8) of cremasteric PCVs stained for CD31 (magenta), MRP-14 (blue), and cortactin (green) from mice injected i.s. with PBS or TNFα for 2 h and mice subjected to AAT (300 ng i.p.) augmentation therapy (AAT + TNFα) (scale bars = 40 μm; $n$ = 3–6 mice/group). Bottom panels show cortactin in 2× zoomed-in orange areas. **(B)** Quantification of cortactin MFI in venular EC using the images from A. **(C)** Quantification of neutrophil transmigration from 1024 × 1024 images from A. **(D–F)** IVM of cremasteric TNFα-inflamed PCVs from mice receiving AAT or saline. Cell flux (D), rolling velocity (E), and adhesion (F) were quantified; $n$ = 3 mice/group. All data are represented as means ± SEM; ∗$P$ < 0.05; ∗∗$P$ < 0.01; ∗∗∗$P$ < 0.001; ∗∗∗∗$P$ < 0.0001.

injection of AAT (Fig. 5, A and B). Concomitantly, TNFα-induced neutrophil extravasation was significantly reduced after AAT treatment (Fig. 5 C). IVM revealed that AAT did not affect neutrophil flux and rolling velocity (Fig. 5, D and E). However, neutrophil adhesion was significantly reduced (Fig. 5 F). This agrees with previous studies where AAT affected $Ca^{2+}$ and LTB4 signaling and integrin expression in neutrophils (McEnery et al., 2022; Al-Omari et al., 2011; O'Dwyer et al., 2015). Whether AAT also regulates these mechanisms in inflamed cremasteric venules remains to be investigated. While neutrophil adhesion and diapedesis are governed by different mechanisms, reduced adhesion commonly precedes reduced diapedesis (León-Vega et al., 2024). Together, the findings identify degradation of endothelial cortactin by NSP as a key element of neutrophil-mediated mechanisms of neutrophil extravasation.

### Relevance of the identified inflammatory endothelial cortactin dynamics

Cortactin regulates endothelial permeability and neutrophil–endothelial interactions via two distinct mechanisms (García Ponce et al., 2016; Schnoor et al., 2011). On the one hand, cortactin deficiency causes increased formation of contractile actin stress fibers via upregulation of the RhoA/ROCK pathway and inhibiting Rap1, leading to endothelial hyperpermeability (García Ponce et al., 2016; Schnoor et al., 2011). On the other

hand, despite the weaker endothelial barrier in the absence of cortactin, neutrophils could not take advantage of the open contacts and instead transmigrated less efficiently through cortactin-deficient EC due to defective ICAM-1 clustering into ring-like structures that neutrophils employ to firmly adhere to EC (Schnoor et al., 2011). These disparate findings could previously not be explained. Guided by the present results, it is tempting to speculate that the total deficiency of cortactin does not comprehensively reveal the physiological role of EC cortactin. Indeed, our detailed analysis of the spatiotemporal dynamics of cortactin suggests a model in which cortactin is needed to establish neutrophil–EC interactions, leading to the transfer of NSP to EC. Transferred NSP will then degrade cortactin in a time-dependent manner and destabilize EC junctions, enabling neutrophils to transmigrate across EC contacts and breach the venular endothelium (see graphical abstract). Critically, the mechanism through which NSP are transferred into EC remains unclear and will require deciphering in future works. This exquisite spatiotemporal regulation of cortactin dynamics by NSP supports efficient neutrophil breaching of the endothelium but may also trigger excessive neutrophil recruitment in pathological settings if not controlled properly. Targeting NSP via AAT to prevent endothelial cortactin degradation may therefore be a novel strategy to fine-tune neutrophil recruitment during inflammatory diseases that are characterized by excessive neutrophil presence and NSP-induced tissue damage.

# Materials and methods

## Reagents and antibodies

For a complete list of all reagents and antibodies used in this study, please see Table S1.

## Mice

WT C57BL/6 mice (RRID:IMSR_JAX:000664) were kept in the animal barrier facility at Cinvestav-IPN under specific pathogen-free conditions and were used at an age range of 8–12 wk and a weight range of 20–25 g. Since all *in vivo* experiments were performed in the cremaster muscle, only male mice were used in this study. All experiments have been approved by the Institutional Animal Care and Use Committee of Cinvestav-IPN and were conducted in accordance with ARRIVE guidelines and local authorities' regulations. Animals were allocated to experimental groups randomly and inspected in a blinded fashion. In all experiments, mice were anesthetized by i.p. injection of ketamine hydrochloride (100 mg/kg) and xylazine (10 mg/kg) and euthanized by anesthesia overdose.

## *In vivo* inflammation models

In anesthetized mice, cremaster muscle inflammation was induced by i.s. injection of either 300 ng TNFα, 50 ng IL-1β, 500 ng CXCL1 (PeproTech), or 300 ng LTB4 (R&D Systems) in 400 μl PBS for periods of 2–4 h. Control mice were injected i.s. with 400 μl of PBS only. All injected solutions contained 4 μl of conjugated anti-CD31 (PECAM-1) antibody clone 390 (Cat #16-0311-85; eBioscience) to label blood vessels.

## Cremaster muscle immunofluorescence staining and confocal microscopy

Cremaster muscles were surgically excised and fixed in 4% PFA for 45 min at 4°C, permeabilized, and blocked in PBS containing 0.5% Triton X-100 and 25% FBS for 4 h at RT with gentle shaking. Then, whole tissues were transferred into microtubes and incubated in 150 μl PBS containing 10% FBS and fluorescently labeled primary antibodies (anti-cortactin-AF488, clone 289H10 [Lai et al., 2009]; anti-MRP-14-AF647, clone 2B10) overnight at 4°C. Tissues were then washed three times with PBS for 15 min at RT. Alternatively, cremaster muscles were incubated with rabbit anti-CatG, followed by secondary goat anti-rabbit-AF647 antibody (Thermo Fisher Scientific). Finally, the whole tissues were washed three times in PBS and mounted on a glass slide with PBS (Joulia et al., 2022). Z-stack images of PCVs of 20–40 μm diameter were acquired using oil immersion 40×/1.25 or 63×/1.4 objectives in a Leica TCS SP8 or oil immersion 60×/1.4 and dry 20×/0.75 objectives in a Nikon A1 confocal laser scanning microscope. To achieve automated super-resolution, 0.35-μm z-stack images of PCVs were acquired using a Leica TCS SP8 confocal laser scanning microscope equipped with a 63× oil-immersion objective and Leica LIGHTNING adaptive deconvolution software (Leica). Analysis of confocal images was performed using Imaris software (Bitplane, RRID:SCR_007370).

## Image analysis of PCVs of cremaster muscle

Image stacks of longitudinal half vessels were reconstructed in 3D and analyzed using Imaris software (Joulia et al., 2022). The MFI values of cortactin and CatG in venular ECs were measured based on the isosurface masking of PECAM1-positive areas representing junctions and cytoplasmic non-junctional regions. To specifically measure CatG internalized by the endothelium, CatG from neutrophils was removed using isosurface masking of MRP14. Then, endothelial CatG was measured using isosurface masking of cytoplasmic and junctional PECAM1-positive areas. 3–8 PCVs per mouse were analyzed. For most of the analysis, all images per mouse were averaged and plotted as the mean per individual mouse per group.

## Neutrophil depletion and flow cytometry of mouse blood

Neutrophils were depleted by i.v. injection of 25 μg rat anti–Gr-1 antibody (Ly6C/Ly6G clone: RB6-8C5; BioLegend) (Hoesel et al., 2005). Control mice were injected i.v. with 25 μg rat IgG2b κ isotype control antibody. 24 h later, local inflammation was induced by i.s. injection of 300 ng TNFα in 400 μl PBS, and cremaster muscles were prepared for confocal microscopy as described above.

After cremaster muscles were dissected and the mice euthanized, blood was obtained by cardiac puncture to confirm neutrophil depletion by flow cytometry. Erythrocytes were lysed from blood samples by adding 1 ml of pre-warmed 1× BD lysing buffer (BD Bioscience) to 100 μl of blood according to the manufacturer's instructions. The cell suspension was washed twice with PBS containing 3% FBS and blocked with ice-cold PBS + 3% FBS containing mouse TruStain FcX (1:100) blocking solution (BioLegend) for 30 min on ice. Staining was performed by incubating cell suspensions in cold PBS + 3% FBS containing APC/Cy7 anti-Ly6G (1:200) and Pacific blue anti-CD45 (1:200)-labeled antibodies for 30 min. Stained cells were fixed by adding 100 μl 4% PFA for 20 min at RT. Cell suspensions were analyzed using a FACS Canto II flow cytometer (Becton Dickinson) and FlowJo Treestar V10 software (RRID:SCR_008520).

## AAT augmentation therapy

60 mg/kg of clinical-grade human AAT (Zemaira, CSL Behring) in 300 μl was injected i.p. into C57BL/6 mice 2 h before i.s. injection of 300 ng TNFα in 400 μl PBS for additional 2 h. The control groups were injected i.p. with 300 μl saline solution for 2 h, followed by i.s. injection of 300 ng TNFα or PBS alone for 2 h. Cremaster muscles were then excised and prepared for immunofluorescence staining as described above. Confocal images were used to quantify the number of transmigrated neutrophils and cortactin degradation using Imaris software.

## Epifluorescence IVM of the cremaster muscle

To investigate the effect of NSP inhibition on the neutrophil extravasation cascade, IVM was performed of cremasteric PCVs of *LysM-EGFP-ki* male mice treated with AAT (Schnoor et al., 2011). 2 hours after AAT therapy as described above, local inflammation was induced by i.s. administration of 300 ng TNFα. 2 hours later, mice were anaesthetized by i.p. injection of anesthetic solution as indicated above. Cremaster muscles were exteriorized and placed using small pins over a custom-built microscope stage. The cremaster muscles were constantly superfused with 37°C warm 1× PBS to maintain the tissue moist

during the whole experiment. PCVs of 20–40 µm diameter were recorded using a 40× water-immersion objective mounted on an intravital upright microscope (Axioscope A1; Carl Zeiss). Rolling and firmly adherent leukocytes were observed by transillumination IVM and epifluorescence microscopy using the GFP filter. Only EGFP-positive neutrophils were considered in the analysis. At least six venules per mouse were recorded. Recorded videos were analyzed using ImageJ software (NIH, Bethesda, MA, USA, RRID:SCR_003070).

## HUVEC

HUVEC were isolated from discarded human umbilical cords and grown in supplemented EC Medium (ScienCell Research Laboratories) containing 10% FBS, 1% penicillin/streptomycin, and 5 µg/ml EC growth supplement. Cells from passages 1–6 were used for experiments. HUVEC were regularly tested negative for mycoplasma contamination. Confluent HUVEC monolayers were stimulated with either 15 ng/ml TNFα, 15 ng/ml IL-1β, or medium alone as control and incubated for 4–24 h in a humidified atmosphere with 5% $CO_2$ at 37°C. Alternatively, HUVEC monolayers incubated with TNFα for 12 h were treated for 30 min with the proteasome inhibitor MG-132 (10–100 µM), the calpain inhibitor ALLN (10–100 µM), the lysosomal inhibitor leupeptin (50–100 µM), or DMSO as vehicle control (Merck).

## Isolation of human blood neutrophils and NSP inhibition

Peripheral blood from consenting healthy donors was collected using BD Vacutainer EDTA K2 tubes (Becton Dickinson). Blood leukocyte populations were separated by Histopaque 1077 and 1119 density gradient (3 ml of solution 1.077 g/ml over 3 ml of 1.119 g/ml solution, Merck) and then centrifuged at 700 × $g$ for 30 min at RT without brakes. Neutrophils were collected from the interface of the Histopaque 1119 and Histopaque 1077 layers and washed twice with 10 ml of ice-cold PBS containing 10% FBS. The remaining erythrocytes were lysed by adding 5 ml cold hypotonic solution (0.2% NaCl, 1% BSA, and 20 mM HEPES) for 1 min on ice. 5 ml of hypertonic solution (1.6% NaCl, 1% BSA, and 20 mM HEPES) was added immediately thereafter. Neutrophils were washed twice with cold PBS and resuspended in cold RPMI-1640 medium (Merck) supplemented with 10% FBS. Isolated neutrophils (1 × $10^6$/ml) were then co-incubated with HUVEC monolayers for 20 min at 37°C. Where indicated, neutrophils were pre-treated with the serine protease inhibitors Siv (ONO-5046, 50–100 µM), PMSF (0.5–2 mM), or AEBSF (0.5 mM) (Sigma-Aldrich) for 30 min at 37°C before transferring to HUVEC monolayers. Control neutrophils were incubated with DMSO or RPMI medium.

## Production of cf-NS

Human primary neutrophils (1 × $10^7$ cells/ml in RPMI medium) were primed with 50 ng/ml TNFα for 20 min at 37°C, followed by 100 ng/ml IL-8 (PeproTech) or 1 µM fMLP stimulation in one well of a 24-well plate pre-coated with 5 µg/ml recombinant hICAM-1-mFc (BioLegend). Neutrophils were allowed to adhere and become activated for an additional 30 min at 37°C. The supernatant was then centrifuged at 400 ×

$g$ for 5 min, followed by 2,000 × $g$ for 10 min at 4°C to precipitate cells and debris. 300 µl of the resulting cf-NS was collected and transferred directly to TNFα-stimulated HUVEC and incubated for 1 h at 37°C. Where indicated, neutrophils were pre-treated with 10 µM NEI20 (Johnson et al., 2016) for 1 h at 37°C before stimulation, or cf-NS were treated with AAT (10 µg/ml) for 30 min at 37°C.

## Flow cytometry analysis of co-cultures

After incubation with either neutrophils or cf-NS, HUVEC monolayers were washed with warm 1× PBS to remove non-adherent neutrophils, detached with TrypLE Express/EDTA (Gibco), and then fixed immediately with 4% PFA for 10 min at RT. After washing twice with cold 1× PBS containing 3% FBS, 0.1% saponin, and 5 mM EDTA, cell suspensions were blocked using human TruStain FcX (1:1,000) (BioLegend) for 20 min at RT. Subsequently, the cells were stained using AF488-labeled mouse anti-cortactin (clone 289H10, 1:200), CD45-PB (1:200) or rabbit anti-CG antibodies (1:200) for 30 min at 4°C. The secondary anti-rabbit-AF-647 antibody (1:1,000) was incubated for 30 min at 4°C. Data were acquired on a FACS Canto II flow cytometer and analyzed using FlowJo Treestar V10 software.

## Incubation of HUVEC with CatG

Confluent HUVEC were incubated with 10 µg/ml human CatG (Abcam) for 1 h at 37°C and washed extensively. Protein extracts were obtained as indicated below and separated by SDS-PAGE. Alternatively, protein extracts from HUVEC monolayers were incubated with 1–2 µg of CatG in Triton X-100 lysis buffer for 20 min on ice immediately after HUVEC lysis. Then lysates were Western blotted as described below.

## Western blot

HUVEC monolayers alone or in co-culture were lysed with Triton X-100 lysis buffer (2% Triton X-100, NaCl 150 mM, Tris-HCl, pH 8.0, 20 mM, $CaCl_2$ 1 mM, leupeptin 15 µg/ml, PMSF 1 mM, and aprotinin 20 µg/ml). Protein extracts were separated by SDS-PAGE and transferred to 0.45-µm-pore nitrocellulose membranes (Bio-Rad). Membranes were blocked in TBS containing 0.1% Tween (TBS-T) and 5% skim milk for 1 h at RT. Then, membranes were incubated with mouse anti-cortactin (clone 289H10), rabbit anti-CatG (Invitrogen), anti-VE-cadherin (clone C-19), γ-tubulin, (Clone GTU-88), Arpin (rabbit polyclonal, homemade [Montoya-Garcia et al., 2024], ArpC5A [Clone 323F2], and rabbit anti-vinculin [clone V4139; Sigma-Aldrich] antibodies or mouse anti-GAPDH [Santa Cruz]) antibody as loading control in blocking solution overnight at 4°C. Blots were washed three times with TBS-T for 10 min before incubation with anti-rabbit or anti-mouse secondary antibodies coupled to horseradish peroxidase (Santa Cruz Biotechnology) for 1 h at RT. Membranes were washed three times with TBS-T, and bands were visualized using SuperSignal West Femto chemiluminescent substrates (Thermo Fisher Scientific) in a ChemiDoc device (Bio-Rad). Pixel intensity was quantified using ImageJ software (NIH, Bethesda, MD, USA).

## Immunofluorescence of HUVEC cultures

After incubation of HUVECs with cf-NS on 0.8% gelatin-coated glass coverslips, cells were fixed with 4% PFA for 10 min at RT and permeabilized with 1× PBS containing 0.1% Triton X-100 + 1% BSA for 10 min at RT. Coverslips were then blocked by washing three times with PBS + 1% BSA for 5 min at RT. Mouse anti-cortactin and rabbit anti-CatG primary antibodies were incubated overnight at 4°C. Then, coverslips were washed three times with PBS and incubated with species-specific, fluorescently labeled anti–rabbit-AF647 and anti–mouse-AF488 secondary antibodies in PBS for 2 h at RT in the dark. Finally, the coverslips were washed three times with PBS and mounted on slides with special ProLong Gold mounting medium containing 4′,6-diamidino-2-phenylindole (DAPI; Thermo Fisher Scientific). Co-cultures were analyzed using a confocal laser scanning microscope Zeiss LSM Airyscan, and z-stack images were taken. Detailed analysis of expression and localization of the indicated molecules was performed using Imaris software.

## Image analysis of HUVEC monolayers using Imaris

CatG internalization in HUVEC was defined as the number of intracellular dots based on isosurface masking of CatG signal using Imaris. Cortactin MFI was measured using isosurface masking of endothelial cortactin signal.

## Flow chambers

$5 \times 10^4$ HUVEC were cultured in P35 petri dishes until 80% confluence. Cells were then stimulated or not with TNFα (15 ng/ml) for 12 h at 37°C and 5% $CO_2$. Neutrophils were isolated as described above. $5 \times 10^5$ neutrophils or whole human peripheral blood were incubated with HUVEC for 15 min at 37°C. The parallel plate flow chamber kit (GlycoTech) was employed for analyzing cell adhesion under flow. Shear stress was applied at 50 dynes/cm² for 10 min with a perfusion pump (Inovenso). Adherent neutrophils were counted under flow using an inverted Axiovert 40C microscope (Zeiss) in six areas of 200 μm² in each condition. Co-cultures were then harvested by trypsinization (0.05% for 5 min) and prepared for flow cytometry as described above.

## Statistics

Statistical analysis was performed using two-tailed Student's *t* test for comparison of two groups or one-way analysis of variance (ANOVA) with Bonferroni's multiple comparisons to compare more than two groups (GraphPad Prism software, RRID:SCR_002798). Data of each experiment are presented as means ± SEM. Values of P < 0.05 were considered statistically significant. Data distribution was assumed to be normal, but this was not formally tested.

## Online supplemental material

Fig. S1 shows cortactin degradation over time during inflammation and that more neutrophil extravasation correlates with lower endothelial cortactin. Fig. S2 shows that cortactin is not degraded by endothelial enzymes. Fig. S3 shows that cortactin is a target for NSP and that TNFα alone does not cause loss of cortactin under flow conditions. Table S1 shows reagents and antibodies used in this study. Video 1 shows the endothelial CatG signal in PCVs of the TNFα-inflamed cremaster using CD31 as template to only visualize the endothelium.

## Data availability

All data are available in the published article and its online supplemental material.

## Acknowledgments

We thank Kasra Askari for his help with HUVEC-neutrophil supernatant co-cultures and immunofluorescence staining. We also thank the flow cytometry core facility at Coordinación de Investigación en Salud, Centro Médico Nacional Siglo XXI, Instituto Mexicano del Seguro Social (IMSS), Mexico City, for providing flow cytometry infrastructure and all veterinarians at the animal facility of Cinvestav-IPN.

This work was funded by a Newton Advanced Fellowship from the Royal Society, UK (NAF/R1/180017), to M. Schnoor and S. Nourshargh. S. Nourshargh was additionally funded by the Wellcome Trust (098291/Z/12/Z and 221699/Z/20/Z). K. Rottner and T.E.B. Stradal acknowledge intramural funding from the Helmholtz Society. This work was also supported by National Institutes of Health (NIH) grant P01HL152958 and R01HL088256 to S.D. Catz; and by IMSS grant R-2022-785-048 to E. Vadillo. Open Access funding provided by CINVESTAV-IPN.

Author contributions: Idaira M. Guerrero-Fonseca: conceptualization, data curation, formal analysis, investigation, methodology, validation, visualization, and writing—original draft, review, and editing. Karina B. Hernández-Almaraz: formal analysis, investigation, validation, and writing—review and editing. Iliana I. León-Vega: formal analysis, investigation, and methodology. Régis Joulia: methodology, resources, and writing—review and editing. Armando Montoya-García: investigation. Hilda Vargas-Robles: investigation and validation. Theresia E.B. Stradal: formal analysis, resources, and writing—review and editing. Klemens Rottner: formal analysis, resources, and writing—review and editing. Reyna Oregon: investigation. Eduardo Vadillo: data curation, formal analysis, investigation, methodology, resources, supervision, visualization, and writing—review and editing. Jennifer L. Johnson: formal analysis, investigation, and methodology. William B. Kiosses: formal analysis, investigation, and visualization. Sergio D. Catz: conceptualization, formal analysis, funding acquisition, investigation, methodology, resources, and writing—review and editing. Sussan Nourshargh: funding acquisition, investigation, methodology, project administration, resources, supervision, validation, and writing—review and editing. Michael Schnoor: conceptualization, funding acquisition, project administration, resources, supervision, validation, visualization, and writing—original draft, review, and editing.

Disclosures: The authors declare no competing interests exist.

Submitted: 3 October 2024

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

## Supplemental material

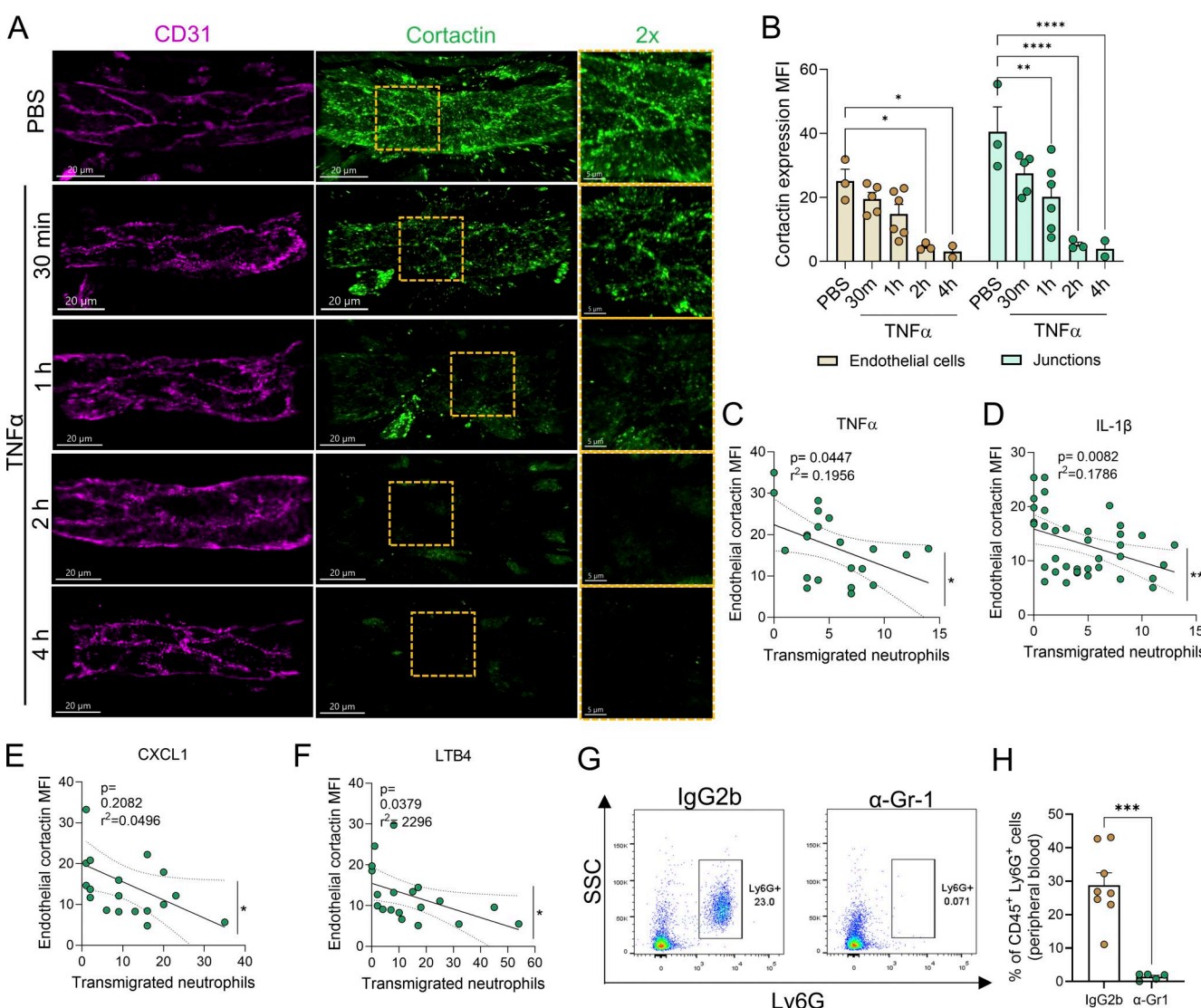

Figure S1. **Neutrophil extravasation correlates with loss of cortactin in** PCV**s over time. (A)** Representative confocal images (Leica SP8) of inflamed cremasteric PCVs (CD31, magenta; cortactin, green) treated with TNFα (300 ng) or PBS for the indicated times (scale bars = 20 μm). Right panels show cortactin (green) in 2× zoomed-in yellow areas; scale bars = 5 μm. **(B)** Quantification of cortactin MFI in venular EC and endothelial junctions from the images shown in A, n = 2–6. **(C–F)** Correlation of the number of extravasated neutrophils and endothelial cortactin MFI in cremaster muscles stimulated with either TNFα (C), IL1-β (D), CXCL1 (E) for 2 h each, or LTB4 for 1 h (F). Each dot represents one PCVs analyzed from 3 to 4 mice per group. Spearman's rank correlation test yielded the p and $r^2$ values shown in each graph. **(G)** Flow cytometry plots of peripheral blood samples from mice subjected to neutrophil depletion (α–Gr-1 antibody) or treated with isotype control antibody (rat IgG2b). The neutrophil population was identified as $CD45^+Ly6G^+SSC^{high}$. **(H)** Quantification of the percentage of neutrophil depletion in peripheral blood 24 h after i.v. injection of isotype control antibody or α–Gr-1 antibody. Data are represented as means ± SEM; n = 3–4 mice/group. ∗P < 0.05; ∗∗P < 0.01; ∗∗∗P < 0.001; ∗∗∗∗P < 0.0001.

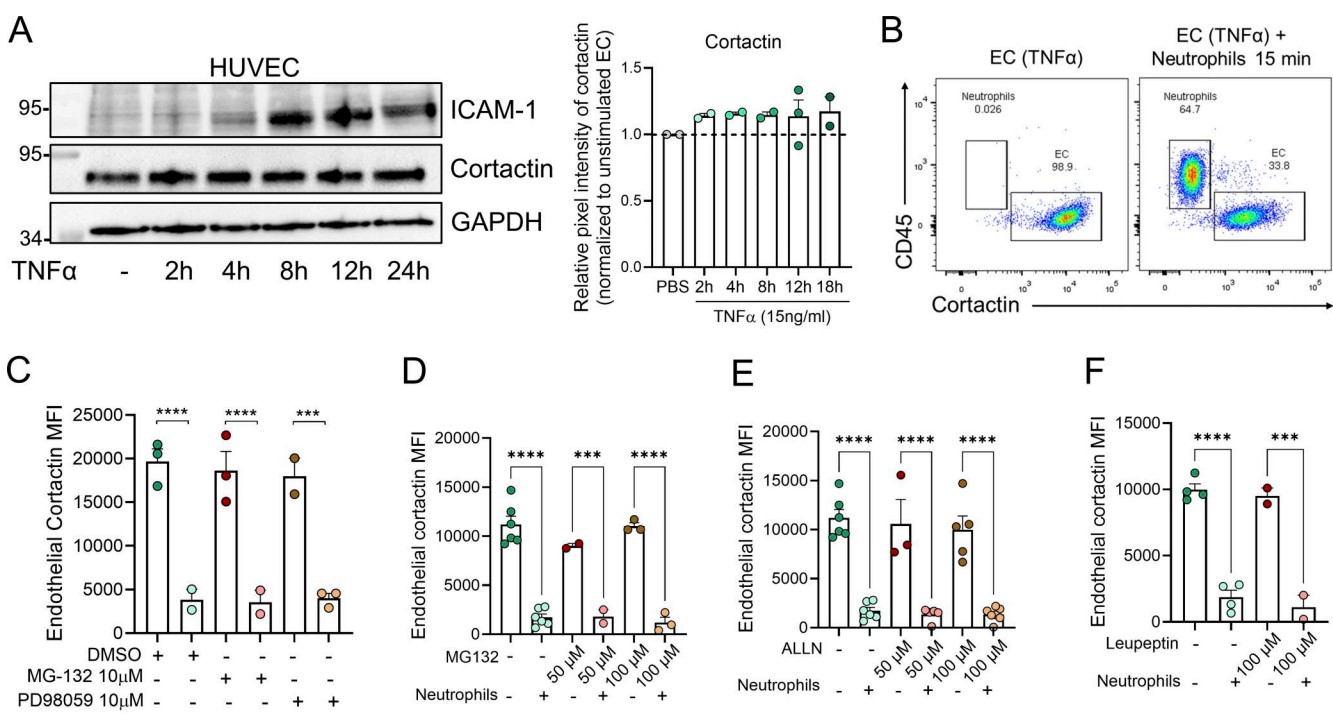

Figure S2.    **Cortactin is not degraded by TNFα stimulation or endothelial enzymes. (A)** Immunoblot of cortactin, ICAM-1, and GAPDH in lysates of HUVEC monolayers stimulated with TNFα (15 ng/ml) for the indicated times. The increase in ICAM-1 shows that the inflammatory stimulus worked. Quantification of cortactin pixel intensities normalized to tubulin expression and unstimulated controls (set to 1, dotted line) is shown in the graph on the right (n = 3). **(B)** Representative flow cytometry dot plots of TNFα-treated HUVEC alone (left, cortactin⁺CD45⁻) and co-cultures with human neutrophils (right, cortactin⁻ CD45⁺) for 20 min **(C–F)** HUVEC monolayers were treated with the proteasome inhibitor MG-132 and the ERK inhibitor PD98059 (C and D), the calpain inhibitor ALLN (E), the lysosome inhibitor leupeptin (F), or DMSO as vehicle control for 2 h before co-culture with neutrophils for 20 min. Cortactin protein levels were then analyzed by flow cytometry and displayed as MFIs; n = 2–4 independent experiments. ∗∗∗P < 0.001; ∗∗∗∗P < 0.0001. Source data are available for this figure: SourceData FS2.

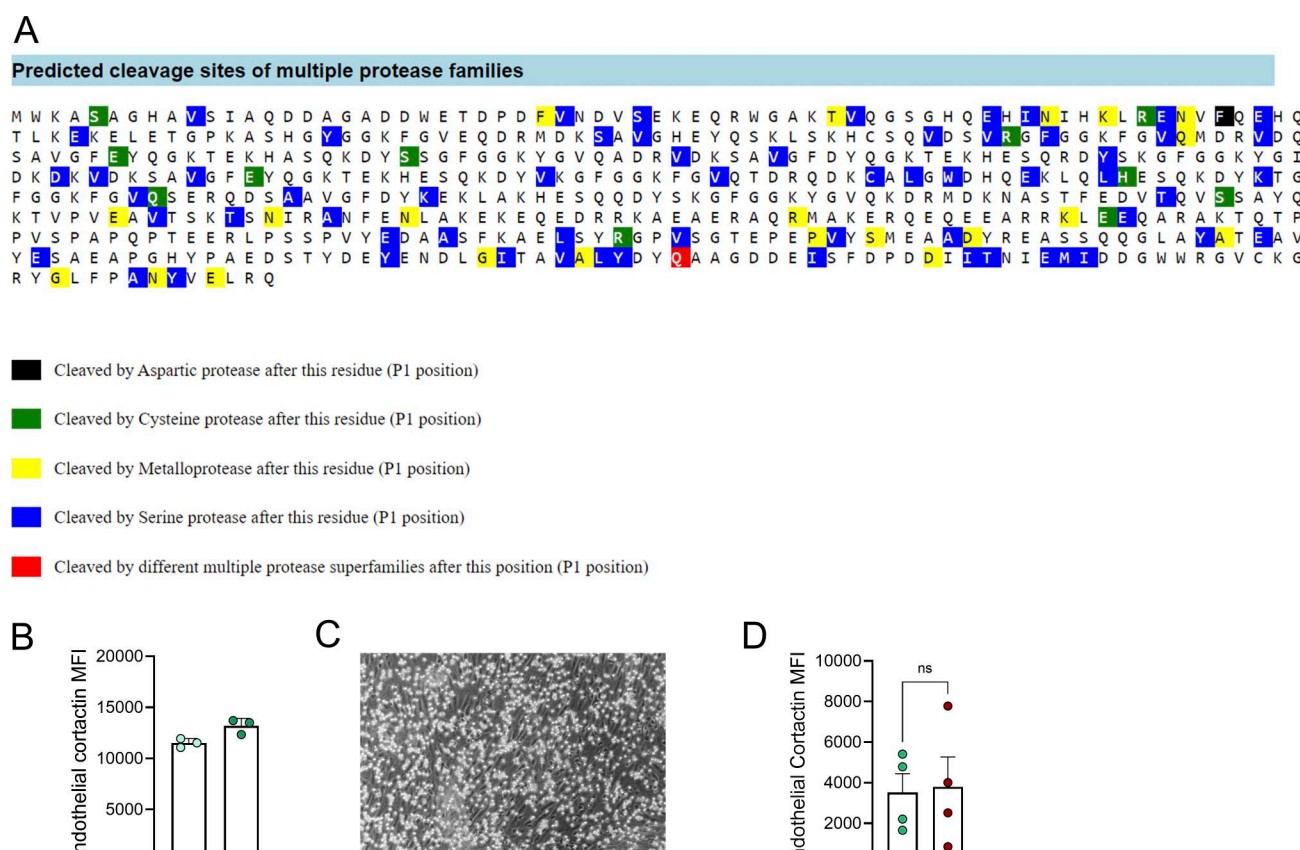

## A

**Predicted cleavage sites of multiple protease families**

Figure S3. **Cortactin is a target for NSP. (A)** Prediction of putative cleavage sites for different protease families in the cortactin sequence using PROSPER software. **(B)** TNFα alone does not induce loss of cortactin under flow conditions; $n = 4$. **(C)** Representative image of neutrophils interacting with HUVEC under flow; $n = 4$; scale bar = 200 µm. **(D)** HUVEC perfusion under flow conditions with whole blood does not further reduce cortactin levels compared with HUVEC perfusion with isolated neutrophils; $n = 4$.

Video 1. **Endothelial CatG in inflamed PCVs.** Rendered isosurface of the CD31-positive endothelial layer of PCVs (cyan) obtained after high-resolution confocal microscopy of TNFα-inflamed cremaster muscle and analysis with Imaris software. CatG within EC (magenta) inside the endothelial CD31 isosurface is shown. The neutrophil staining and neutrophil CatG are omitted for clear visualization of CatG in the endothelial plane.

**Provided online is Table S1. Table S1 shows reagents and antibodies used in this study.**

