## [Peer Review File · The Journal of Cell Biology]

Neutrophil serine proteases degrade endothelial cortactin and promote extravasation

Idaira Guerrero-Fonseca, Karina Hernández-Almaraz, Iliana León-Vega, Régis Joulia, Armando Montoya-García, Hilda Vargas-Robles, Theresia Stradal, Klemens Rottner, Reyna Oregon, Eduardo Vadillo, Jennifer Johnson, William Kiosses, Sergio Catz, Sussan Nourshargh, and Michael Schnoor

Corresponding Author(s): Michael Schnoor, Center for Research and Advanced Studies of the National Polytechnic Institute

Review Timeline:

Submission Date:	2024-10-03
Editorial Decision:	2024-12-04
Revision Received:	2026-03-05
Editorial Decision:	2026-03-20
Revision Received:	2026-03-27

Monitoring Editor: Anna Huttenlocher

Scientific Editor: Dan Simon

Transaction Report:

DOI: <https://doi.org/10.1083/jcb.202410019>

December 4, 2024

Re: JCB manuscript #202410019

Michael Schnoor
Center for Research and Advanced Studies of the National Polytechnic Institute

Dear Dr. Schnoor,

Thank you for submitting your manuscript entitled "Neutrophil extravasation requires endothelial cortactin degradation by neutrophil serine proteases." The manuscript has been evaluated by expert reviewers, whose reports are appended below. Unfortunately, after an assessment of the reviewer feedback, our editorial decision is against publication in JCB.

You will see that the reviewers find the premise of your study interesting but also state that the data does not convincingly demonstrate that cathepsin G released from neutrophils is internalized by endothelial cells and then specifically degrades cortactin. They also note that the mechanism by which cathepsin G is transferred from neutrophils into endothelial cells is unclear.

While your manuscript is intriguing, we feel that addressing the reviewer comments would require a more substantial amount of new experiments than can be done in a typical revision period. If you wish to expedite publication of the current data, it may be best to pursue publication at another journal.

However, given interest in the topic, we would be open to resubmission to JCB of a significantly revised manuscript that provides stronger evidence for the main claims regarding cathepsin G internalization by endothelial cells and specific effects on cortactin. As stated by the reviewers, this would necessitate higher resolution microscopy. For a Report a detailed mechanistic investigation of cathepsin G release and transfer would not be required. However, if you do carry out such experiments it would make this a stronger and more compelling study that would then be suitable for our Article format.

Regardless of how you choose to proceed, we hope that the comments below will prove constructive as your work progresses. We would be happy to discuss the reviewer comments further once you've had a chance to consider the points raised in this letter. You can contact the journal office with any questions at cellbio@rockefeller.edu.

Thank you for thinking of JCB as an appropriate place to publish your work.

Sincerely,

Anna Huttenlocher, MD, PhD
Monitoring Editor
Journal of Cell Biology

Dan Simon, PhD
Scientific Editor
Journal of Cell Biology

Reviewer #1 (Comments to the Authors (Required)):

The paper by Guerrero-Fonseca proposes that neutrophil adhesion to activated endothelium triggers the release (by neutrophils) and subsequent uptake (by EC) of proteases that degrade junctional cortactin, in order to allow efficient neutrophil TEM. This is an interesting concept and study, which also raises several questions.

The work heavily relies on imaging and quantification of immunofluorescent signals. While part of the data, both in vitro and in vivo, are suggestive indeed that cortactin is regulated in some way at sites of neutrophil adhesion, there are several technical aspects of the study that are not entirely convincing and there are a number of questions relevant to the model that are not sufficiently addressed.

For example, is this mechanism specific to cortactin? It would have made sense if the authors would have included specificity controls such as filamin or alpha-actinin in some of the key experiments. Similarly, why would this only target junctional, and not apical, cortactin? The association of cortactin to a cell adhesion molecule which is interacting with the subcortical actin cytoskeleton is likely not so different between the Icam-1 associated pool of cortactin vs the VE-cad associated pool. And if it is,

then this should have been addressed.

Another question is why the junctional CD31 staining was more or less similar in situations with and without cortactin - this is not in line with the model which predicts loss of integrity upon loss of cortactin.

Technically, I find that the imaging, although in part of good quality, does not allow to reach the conclusions that the authors draw from their data. To begin with, the loss of cortactin immunostaining is directly interpreted as degradation, in the absence of any evidence that this indeed occurs. For example, one cannot exclude that by some mechanism (binding to nearby proteins, post translational modifications) the detection of cortactin becomes less efficient. In any case, there are no control experiments done to address this.

Along the same line, there is no biochemical approach to detect the degradation in different way, which would have made the model more convincing. Also - how about live imaging of a FP-tagged cortactin? This would have controlled for antibody detection issues and shown if perhaps translocation is at play here.

In Fig 2, near complete loss of cortactin staining is shown at sites of neutrophil accumulation, while the model predicts that the apical cortactin would still be visible, so how do the authors explain this?

In Fig 3A, there is already a substantial loss of cortactin induced by neutrophils in non-TNF treated endothelial cells. This is unexpected as there should be no or very little interaction with the endothelium in untreated control conditions.

The monocyte control experiment (3C) is not convincing as it is not shown if the adhesion of the MNC was as efficient as for the neutrophils.

Fig 3D should be complemented with imaging of additional markers (VE cad, F-actin etc) showing that the same optical section is shown, in particular as docking structures are usually more clear when imaged on top of the EC body.

The use of cellular extracts to study protein degradation is (as in fig S3H), in particular when working with neutrophils, not very convincing. These cells are full of proteolytic activity which may lead to incorrect conclusions, as happened before in the field of leukocyte TEM many years ago. Moreover, fig S3H should have been complemented by a specificity control, which unfortunately is lacking.

In Fig 4A, the imaging data are not strong enough to conclude that CG is indeed inside and not on the EC. These cells are so thin in relation to the resolution of the imaging used here, that this is not convincing. Additional superresolution imaging or alternative means to show that CG has been internalized would be required.

Similarly, fig 4F is hard to judge - it's not entirely clear to this reviewer what is visualized and on what basis the authors conclude 'that a significant amount of CG was associated with ECs' as there is also a lot of green staining outside the vessel

In 5F that AAT condition shows much less neutrophils and the text suggests that these and the many neutrophils in the middle, TNF image represent extravasated cells. How can the reader see this? Moreover, if there is less TEM, one would expect more adhesive neutrophils being retained on the TNF activated vessel, even more so as the model predicts that docking and ICAM1 function is not altered.

So overall, this study appears potentially interesting but also preliminary, as many key aspects (which protease is essential, selectivity of (junctional) cortactin targeting, mechanism of entry of the protease, etc) are not resolved, and part of the data are not yet convincing enough.

Reviewer #2 (Comments to the Authors (Required)):

The authors think that "EC cortactin is proteolytically degraded by neutrophil serine proteases (NSP) during neutrophil-endothelial interactions". This raises the question how these large enzymes get into ECs. It seems like cortactin dynamics separate permeability (histamine leaves cortactin intact) from neutrophil transmigration (TNF and others lead to cortactin degradation). This study is very interesting and the reported (but invalid) findings would be novel. The interpretation and therefore the proposed mechanisms are not supported by the data.

1. Endothelial cells are 0.1 μm thin in the body and about 1 μm over the nucleus. Thus, it is not possible to distinguish intracellular from extracellular CG and other proteases using confocal microscopy (not enough resolution). The isosurfaces provided by Imaris cannot overcome this deficiency. The term "EC cortactin" is misleading. The fundamental physics of diffraction-limited imaging do not allow this conclusion. This problem can be overcome by superresolution microscopy like STED or STORM.

2. Because of the limited resolution, the authors have no valid evidence that neutrophil GC enters into ECs.

3. In the graphical abstract, the ECs are depicted as being thicker than the neutrophil. This is highly misleading.

4. That permeability and neutrophil transmigration are independently regulated has been known since the late 1980's. Please cite the prior evidence.

5. The authors used the inhibitors PMSF and AEBSF. Evidence for specificity must be provided. Negative controls are needed (structurally similar but inactive molecules). Same for AAT.

6. Then, the authors focus on cathepsin G (CG), which is one of the possible proteases. Cathepsin G knockout mice would

definitively show whether it is CG or some other NSP protease.

7. The transfer mechanism appears to be independent of neutrophil cellular activity. This seems to contradict the observation that CG-containing azurophilic granules are exocytosed from neutrophils in close proximity to ECs. The authors fail to identify the mechanism by which CG is taken up into ECs. They speculate that this "could involve uptake of NSP-containing granules via receptor-mediated endocytosis, micropinocytosis, or nanotubes". However, all of these are active cellular mechanisms and inconsistent with the observed effects of neutrophil extracts with no live neutrophils present.

8. Nexinhib 20 is not specific for azurophilic granules. Better controls are needed.

9. The microscopic MFI will change with time by photobleaching. Thus, MFIs obtained at different times will differ (without any biological reason).

10. The neutrophil depletion experiment uses anti-GR1, which binds Ly-6C and Ly-6G. The neutrophils (after depletion) were detected by anti-Ly-6G. This method is valid only if the authors can show that anti-GR1 does not block anti-Ly6G binding. If it does, the neutrophils are still circulating but become invisible to this method of detection.

Reviewer #3 (Comments to the Authors (Required)):

Guerrero-Fonseca and colleagues describe in their elegant study how endothelial cell cortactin degradation contributes to neutrophil recruitment during inflammation using the mouse cremaster muscle model and additional HUVEC-based in vitro approaches. In a first step, they show that cremaster muscle postcapillary venule endothelial cells lose cortactin expression under in vivo conditions when stimulated with TNF, IL1b, CXCL1 or LTB4 (but not with histamine!). Interestingly, loss of cortactin was dependent on the presence of neutrophils as depletion of neutrophils preserved cortactin expression in inflamed endothelial cells. To demonstrate the critical role of neutrophils for EC cortactin loss, the authors used HUVECs stimulated with TNF +/- neutrophils and found that the presence of neutrophils alone was sufficient to induce EC cortactin loss. Of note, co-culture of HUVECs with monocytes was not able to induce cortactin loss. To explore whether neutrophil specific proteases (NSP) might be involved in neutrophil-dependent EC cortactin loss, the authors used a NE inhibitor as well as non-specific NSPs. While NE inhibition did not show any effect on cortactin loss, NSP inhibition strongly abrogated cortactin loss in neutrophil-stimulated HUVECs suggesting that neutrophil-released NSPs might be taken up by HUVECs and lead to degradation of cortactin. Next, the authors explored whether the serine protease cathepsin G (CG) might be involved in cortactin degradation. Indeed, the authors could detect CG in neutrophil-stimulated HUVECs. Furthermore, using the supernatant of stimulated neutrophils and incubate the supernatant with stimulated HUVECs led to EC CG detection as well as to reduced cortactin levels. These results indicate that CG might be involved in cortactin degradation. To further explore whether released CG come from azurophilic granule secretion, Rab27 dependent azurophilic granule mobilization and release was blocked with NEI20. This led to a partial reduction in EC cortactin degradation suggesting that other mechanisms might be involved in CG release from neutrophils. Alternatively, other NSPs might be involved as well.

Overall, the ms is well written and the results are quite interesting. Please find below some major and minor issues, which should be addressed by the authors. If the authors are able to address these points, the revised ms could also be submitted as full-length article, in my opinion.

- 1) You might want to change the title. You did not conclusively show that neutrophil extravasation is dependent on cortactin loss. You only demonstrate that there is a negative correlation btw EC cortactin and neutrophil extravasation. Focus of the ms is on EC cortactin and CG-dependent cortactin degradation.
- 2) One major component missing is what signal triggers the release of cathepsin G or other NSP from neutrophils during the recruitment process. This could be a local signal triggered through interaction of neutrophils with endothelial adhesion molecules (f.e. E-selectin, ICAMs) or chemokines presented on the inflamed endothelium. The authors could address this in the cremaster muscle using respective blocking abs and f.e. TNF as local stimulus and then look for cortactin loss.
- 2) Along the same line, systemic components in the circulation might also be involved to trigger CG release. Local vs. systemic could be further addressed in flow chambers using stimulated HUVECs and whole blood. Perfusion of the chambers with whole blood for 30 min and subsequent staining of HUVEC cortactin might help to clarify this. Again, you could use blocking antibodies to see what adhesion molecules are involved in triggering cortactin loss. Did you check for systemic CG levels in vivo?
- 3) In my opinion, the static in vitro approach with HUVEC and isolated neutrophils used to confirm the in vivo results from the cremaster muscle might very likely be based on different mechanisms concerning the release of cathepsin G. Here, cell death with passive release of CG could also contribute to the effect. Flow chamber experiments as described in 2) might help to clarify this.
- 3) All figures: please add dots from single experiments to all bar graphs.

Dear Dr. Schnoor,

Thank you for submitting your manuscript entitled "Neutrophil extravasation requires endothelial cortactin degradation by neutrophil serine proteases." The manuscript has been evaluated by expert reviewers, whose reports are appended below. Unfortunately, after an assessment of the reviewer feedback, our editorial decision is against publication in JCB. You will see that the reviewers find the premise of your study interesting but also state that the data does not convincingly demonstrate that cathepsin G released from neutrophils is internalized by endothelial cells and then specifically degrades cortactin. They also note that the mechanism by which cathepsin G is transferred from neutrophils into endothelial cells is unclear.

While your manuscript is intriguing, we feel that addressing the reviewer comments would require a more substantial amount of new experiments than can be done in a typical revision period. If you wish to expedite publication of the current data, it may be best to pursue publication at another journal.

However, given interest in the topic, we would be open to resubmission to JCB of a significantly revised manuscript that **provides stronger evidence for the main claims regarding cathepsin G internalization by endothelial cells and specific effects on cortactin**. As stated by the reviewers, **this would necessitate higher resolution microscopy**. For a Report a detailed mechanistic investigation of cathepsin G release and transfer would not be required. However, if you do carry out such experiments it would make this a stronger and more compelling study that would then be suitable for our Article format. Regardless of how you choose to proceed, we hope that the comments below will prove constructive as your work progresses. We would be happy to discuss the reviewer comments further once you've had a chance to consider the points raised in this letter. You can contact the journal office with any questions at cellbio@rockefeller.edu.

Thank you for thinking of JCB as an appropriate place to publish your work.

Sincerely,

Anna Huttenlocher, MD, PhD

Dan Simon, PhD

Author's Response: Thank you for considering a resubmission. We have now provided much stronger evidence by super-resolution imaging for the presence of cathepsin G inside of endothelial cells in cultured endothelium and in cremaster venules. Moreover, we have performed flow-chamber assays as requested by reviewer 3 that confirmed our static co-culture experiments. Furthermore, we provide evidence for specific degradation of cortactin as other actin-related proteins (vinculin, arpin, Arp2/3 complex) were not seen to be degraded. We are confident that the significant additional new data strengthen our conclusions and hope that the manuscript is now considered acceptable for publication as a JCB report. In this context, we appreciate the editorial guidance that "For a Report a detailed mechanistic investigation of cathepsin G release and transfer would not be required." While we are eager to perform these experiments in the future, unfortunately, our current resources do not allow us to perform these experiments, and therefore we would be grateful for consideration of our revised resubmitted manuscript as a Report.

Reviewer #1 (Comments to the Authors (Required)):

The paper by Guerrero-Fonseca proposes that neutrophil adhesion to activated endothelium triggers the release (by neutrophils) and subsequent uptake (by EC) of proteases that degrade junctional cortactin, in order to allow efficient neutrophil TEM. This is an interesting concept and study, which also raises several questions.

The work heavily relies on imaging and quantification of immunofluorescent signals. While part of the data, both in vitro and in vivo, are suggestive indeed that cortactin is regulated in some way at sites of neutrophil adhesion, there are several technical aspects of the study that are not entirely convincing and there are a number of questions relevant to the model that are not sufficiently addressed.

For example, is this mechanism specific to cortactin? It would have made sense if the authors would have included specificity controls such as filamin or alpha-actinin in some of the key experiments. Similarly, why would this only target junctional, and not apical, cortactin? The association of cortactin to a cell adhesion molecule which is interacting with the subcortical actin cytoskeleton is likely not so different between the Icam-1 associated pool of cortactin vs the VE-cad associated pool. And if it is, then this should have been addressed.

Another question is why the junctional CD31 staining was more or less similar in situations with and without cortactin - this is not in line with the model which predicts loss of integrity upon loss of cortactin.

Authors' response: Thanks for raising these concerns. Our new data show that the effect is indeed specific for cortactin because other actin-related proteins including vinculin, arpin, and Arp2/3 are not degraded (new figure 3G). Another important fact proven by this experiment is that the effect is not an unspecific artifact due to neutrophil fixation and lysis (which is indeed always a concern when working with neutrophils).

We removed the preliminary data on cortactin pools (interacting with either VE-cadherin or ICAM-1) as this claim was not the key message and we currently do not have the means of analyzing this finding in a more convincing fashion.

While CD31 is an important adhesion molecule, it is not the most important molecule regulating permeability. We have previously shown that CD31 as well as other junction proteins are not affected by cortactin deficiency, but that permeability effects in the absence of cortactin are rather a consequence of increased actomyosin contractility (see García-Ponce et al. Sci. Rep. 2016). This is discussed on pages 15-16.

Technically, I find that the imaging, although in part of good quality, does not allow to reach the conclusions that the authors draw from their data. To begin with, the loss of cortactin immunostaining is directly interpreted as degradation, in the absence of any evidence that this indeed occurs. For example, one cannot exclude that by some mechanism (binding to nearby proteins, post translational modifications) the detection of cortactin becomes less efficient. In any case, there are no control experiments done to address this. Along the same line, there is no biochemical approach to detect the degradation in different way, which would have made the model more convincing. Also - how about live imaging of a FP-tagged cortactin? This would have controlled for antibody detection issues and shown if perhaps translocation is at play here.

In Fig 2, near complete loss of cortactin staining is shown at sites of neutrophil accumulation, while the model predicts that the apical cortactin would still be visible, so how do the authors explain this?

Authors' response: We thank the reviewer for this comment, which prompted us to perform super-resolution imaging with cremaster muscles that provided more convincing imaging data that clearly demonstrate the presence of cathepsin G inside endothelial cells of the venules where cortactin is lost (new figure 4). While STED and STORM systems are currently not available to us in Mexico, we used Zeiss LSM Airyscan (Link: ZEISS Airyscan | Super-resolution imaging and molecular measurements) or Leica Lightning (Link: Obtain Maximum Information from your Specimen with LIGHTNING | Learn & Share | Leica Microsystems) systems that reach super-resolution of 90 and 120 nm, respectively, that was sufficient to resolve the CatG signals inside endothelial cells.

The new flow chamber experiments, in which we compared by flow cytometry permeabilized vs non-permeabilized HUVEC after coculture with neutrophils, also confirmed that cathepsinG is only detected after permeabilization inside HUVEC and not on the surface in non-permeabilized HUVEC.

We speak now of degradation only after having confirmed that the loss of signal in the imaging experiments was confirmed by Western blot, where the cortactin band also disappeared. In WB, protein/antibody interaction effects as in IF are not a problem.

Regarding differences in junctional vs apical staining in cells and tissue, we have now removed these statements from the manuscript as we currently cannot provide more convincing experimental evidence as stated above. We have rewritten the text accordingly and hope to address this finding in more detail in the future.

While using FP-tagged cortactin expression in the endothelial cells would have been another independent proof of concept, it would have introduced a new problem as the fluorescent protein could mask the cleavage site. Therefore, we decided to not perform such experiments and hope that the new data and text modifications are convincing enough.

In Fig 3A, there is already a substantial loss of cortactin induced by neutrophils in non-TNF treated endothelial cells. This is unexpected as there should be no or very little interaction with the endothelium in untreated control conditions.

The monocyte control experiment (3C) is not convincing as it is not shown if the adhesion of the MNC was as efficient as for the neutrophils.

Fig 3D should be complemented with imaging of additional markers (VE cad, F-actin etc) showing that the same optical section is shown, in particular as docking structures are usually more clear when imaged on top of the EC body.

Authors' response: Cultured endothelial cells including HUVEC do express a low basal amount of selectins and ICAM1 in untreated conditions that is sufficient to allow for interactions with neutrophils (see Schnoor et al. JEM, 2011, supplemental figures). Also, as this is a static experiment, neutrophils sink to the endothelial surface and have ample

unspecific contact with the apical endothelial side that seems to be sufficient to induce interaction and NSP transfer. However, to further address this point and as suggested by reviewer 3, we performed more physiological flow chamber experiments that confirmed cortactin degradation and cathepsin transfer (Figure 3 I-J), and showed that under flow conditions TNF pretreatment is more important for cortactin degradation. In our static experiments, PMN and MNC interaction with HUVEC is comparable. Anyway, in our new flow chamber experiment, we performed experiments with isolated neutrophils and whole blood showing that cortactin degradation is similar indicating that other cells in the blood (MNC) do not significantly contribute to the degradation effect (new figure S3D).

As discussed above, we have removed old Fig 3D and related claims, and instead focus more on the novel effect that neutrophil NSP exert on intracellular endothelial proteins in vivo as a novel concept in vascular biology, for which the evidence is now much stronger with our new cremaster superresolution imaging data (new Figure 4).

The use of cellular extracts to study protein degradation is (as in fig S3H), in particular when working with neutrophils, not very convincing. These cells are full of proteolytic activity which may lead to incorrect conclusions, as happened before in the field of leukocyte TEM many years ago. Moreover, fig S3H should have been complemented by a specificity control, which unfortunately is lacking.

Authors' response: We are aware of this problem. This experiment was only to demonstrate that neutrophil extracts are indeed able to degrade cortactin to pave the way for the following experiments. GAPDH used as negative control is not degraded. Importantly, our new WB data show that vinculin, arpin, and Arp2/3 are not degraded (new figure 3G), so that we are not dealing with a neutrophil-driven unspecific degradation artefact. Neutrophil supernatants (no extracts involved) and NSP inhibitors used as controls confirmed the data and proved specificity (Figures 3E-G). Thus, we are certain that the observed effects are specific for released neutrophil NSPs and not a general neutrophil lysis problem causing unspecific degradation as was a problem in previous studies as correctly pointed out.

In Fig 4A, the imaging data are not strong enough to conclude that CG is indeed inside and not on the EC. These cells are so thin in relation to the resolution of the imaging used here, that this is not convincing. Additional superresolution imaging or alternative means to show that CG has been internalized would be required.

Similarly, fig 4F is hard to judge - it's not entirely clear to this reviewer what is visualized and on what basis the authors conclude 'that a significant amount of CG was associated with ECs' as there is also a lot of green staining outside the vessel

Authors' response: We thank the reviewer for this comment, which prompted us to perform super-resolution imaging with cremaster muscles that provided more convincing imaging data and that clearly demonstrate the presence of cathepsin G inside endothelial cells of the venules (new figure 4).

In 5F that AAT condition shows much less neutrophils and the text suggests that these and the many neutrophils in the middle, TNF image represent extravasated cells. How can the reader see this? Moreover, if there is less TEM, one would expect more adhesive neutrophils being retained on the TNF activated vessel, even more so as the model predicts that docking and ICAM1 function is not altered.

Authors' response: Apologies for not being clear. The red CD31 staining delineates the venules, and everything outside the red staining are extravasated neutrophils within the inflamed interstitial tissue stained in blue for MRP-14. Such cremaster stainings are common practice to determine and quantify neutrophil extravasation. We now also performed intravital microscopy of the inflamed cremaster and found that AAT indeed inhibits adhesion in vivo (new figure 5F), which was a highly significant and expected finding in agreement with previous studies as discussed now on page 15. It is normal that reduced adhesion precedes reduced extravasation because firm adhesion is a prerequisite to prepare the neutrophil for transmigration as has been shown in many other studies. Without firm adhesion the neutrophils will eventually simply be washed away by the shear force of blood flow.

Reviewer #2 (Comments to the Authors (Required)):

The authors think that "EC cortactin is proteolytically degraded by neutrophil serine proteases (NSP) during neutrophil-endothelial interactions". This raises the question how these large enzymes get into ECs. It seems like cortactin dynamics separate permeability (histamine leaves cortactin intact) from neutrophil transmigration (TNF and others lead to cortactin degradation). This study is very interesting and the reported (but invalid) findings would be novel. The interpretation and therefore the proposed mechanisms are not supported by the data.

1. Endothelial cells are 0.1 μm thin in the body and about 1 μm over the nucleus. Thus, it is not possible to distinguish intracellular from extracellular CG and other proteases using confocal microscopy (not enough resolution). The isosurfaces provided by Imaris cannot overcome this deficiency. The term "EC cortactin" is misleading. The fundamental physics of diffraction-limited imaging do not allow this conclusion. This problem can be overcome by superresolution microscopy like STED or STORM.

Authors' response: We thank the reviewer for this comment, which prompted us to perform super-resolution imaging with HUVEC and cremaster muscles that provided more convincing imaging data and that clearly demonstrate the presence of cathepsin G inside endothelial cells (new figure 4). Presence of cathepsin G inside endothelial cells has been independently proven by flow chamber and flow cytometry assays in which cathepsin G in HUVEC was only detected after permeabilization (see responses to reviewer 3 and new figures 3I-J).

While STED and STORM systems are currently not available to us in Mexico, we used Zeiss LSM Airyscan (Link: ZEISS Airyscan | Super-resolution imaging and molecular measurements)

or Leica Lightning (Link: Obtain Maximum Information from your Specimen with LIGHTNING | Learn & Share | Leica Microsystems) systems that reach super-resolution of 90 and 120 nm, respectively, that was sufficient to resolve the CatG signals inside endothelial cells.

2. Because of the limited resolution, the authors have no valid evidence that neutrophil GC enters into ECs.

Authors' response: Our new super-resolution imaging data now clearly demonstrate the presence of cathepsin G inside endothelial cells (see new figure 4).

3. In the graphical abstract, the ECs are depicted as being thicker than the neutrophil. This is highly misleading.

Authors' response: Thank you. It is difficult to draw such cartoons to scale and fit all required information. The graphical abstract has been adapted, and the statement "Objects are not drawn to scale" has been added.

4. That permeability and neutrophil transmigration are independently regulated has been known since the late 1980's. Please cite the prior evidence.

Authors' response: This has been updated accordingly, and the following citations have been added PMID: 9572834, PMID: 2842887, PMID: 9626051, PMID: 13955841.

5. The authors used the inhibitors PMSF and AEBSF. Evidence for specificity must be provided. Negative controls are needed (structurally similar but inactive molecules). Same for AAT.

Authors' response: Unfortunately, we do not have access to structurally similar but inactive versions of PMSF, AEBSF, or AAT. However, given that all three NSP inhibitors showed similar results, while other inhibitors did not have effects (Suppl Fig S2 C-F, Fig 3A (Sivelestat)), this robust pharmacological strategy strongly suggests a role for NSP.

6. Then, the authors focus on cathepsin G (CG), which is one of the possible proteases. Cathepsin G knockout mice would definitively show whether it is CG or some other NSP protease.

Authors' response: We agree that experiments using cathepsin G KO mice would greatly contribute to clarifying some mechanistic aspects of the manuscript. However, we do not have access to these mice in Mexico and importation restrictions would delay such experiments significantly. Thus, this is currently not an option. However, we do not claim that the effect is CatG specific and in fact believe that multiple NSP can be transferred to ECs and target cortactin. For example, the use of sivelestat (specific for NE, Fig. 3A) alone did not inhibit cortactin degradation. Thus, in the presence of sivelestat cortactin is still degraded, likely by CatG or PR3. Thus, using a mouse deficient for only one NSP may not recapitulate what we are seeing here; a triple KO (cathepsin G, proteinase 3 and elastase) will most likely be required, which does not exist to the best of our knowledge.

7. The transfer mechanism appears to be independent of neutrophil cellular activity. This seems to contradict the observation that CG-containing azurophilic granules are exocytosed from neutrophils in close proximity to ECs. The authors fail to identify the mechanism by which CG is taken up into ECs. They speculate that this "could involve uptake of NSP-containing granules via receptor-mediated endocytosis, micropinocytosis, or nanotubes". However, all of these are active cellular mechanisms and inconsistent with the observed effects of neutrophil extracts with no live neutrophils present.

Authors' response: We would be more than happy to provide the exact transfer mechanism, but, unfortunately, at present we do not have the resources to perform such mechanistic experiments. However, our experiments using Nexinhib20 (exocytosis inhibitor) suggest that neutrophils release extracellular vesicles that are taken up by endothelial cells. Nexinhib abolished cortactin degradation when using supernatants from activated neutrophils that are known to contain vesicles. Also, supernatants of nonactivated neutrophils (containing none or few vesicles) did not degrade cortactin (Figure 4). Also, we have shown that recombinant cathepsinG added to intact HUVEC is not taken up and does not degrade cortactin (Fig 3D). We have discussed this in more detail on page 14.

As the manuscript is being re-submitted as a Report, and the editors did not request for the transfer mechanism in a Report, we hope to address detailed associated mechanisms in future works.

8. Nexinhib 20 is not specific for azurophilic granules. Better controls are needed.

Authors' response: True, and we do not claim this to be the case. The purpose of this experiment was to demonstrate that it is a released neutrophil product that induces cortactin degradation as Nexinhib20 inhibits exocytosis. Nevertheless, the experiments using supernatants (Figures 3E-H) and Nexinhib (Figures 4A-D) demonstrate that it is indeed a secreted neutrophil product that causes cortactin degradation.

9. The microscopic MFI will change with time by photobleaching. Thus, MFIs obtained at different times will differ (without any biological reason).

Authors' response: True; and therefore, all experiments have been done using the same conditions and timing. All tissues have been scanned at equal conditions and within the same timeframe, so that all images are comparable.

10. The neutrophil depletion experiment uses anti-GR1, which binds Ly-6C and Ly-6G. The neutrophils (after depletion) were detected by anti-Ly-6G. This method is valid only if the authors can show that anti-GR1 does not block anti-Ly6G binding. If it does, the neutrophils are still circulating but become invisible to this method of detection.

Authors' response: Antibodies against Gr-1 have been used in multiple studies to deplete neutrophils and proven to deplete more than 90% of the neutrophil population. An antibody-independent detection of neutrophils is provided in the confocal images, where neutrophils

are identified by DAPI instead of Ly6G confirming the absence of neutrophils in TNF-inflamed venules (Figure 2C).

Reviewer #3 (Comments to the Authors (Required)):

Guerrero-Fonseca and colleagues describe in their elegant study how endothelial cell cortactin degradation contributes to neutrophil recruitment during inflammation using the mouse cremaster muscle model and additional HUVEC-based in vitro approaches. In a first step, they show that cremaster muscle postcapillary venule endothelial cells lose cortactin expression under in vivo conditions when stimulated with TNF, IL1b, CXCL1 or LTB4 (but not with histamine!). Interestingly, loss of cortactin was dependent on the presence of neutrophils as depletion of neutrophils preserved cortactin expression in inflamed endothelial cells. To demonstrate the critical role of neutrophils for EC cortactin loss, the authors used HUVECs stimulated with TNF +/- neutrophils and found that the presence of neutrophils alone was sufficient to induce EC cortactin loss. Of note, co-culture of HUVECs with monocytes was not able to induce cortactin loss. To explore whether neutrophil specific proteases (NSP) might be involved in neutrophil-dependent EC cortactin loss, the authors used a NE inhibitor as well as non-specific NSPs. While NE inhibition did not show any effect on cortactin loss, NSP inhibition strongly abrogated cortactin loss in neutrophil-stimulated HUVECs suggesting that neutrophil-released NSPs might be taken up by HUVECs and lead to degradation of cortactin. Next, the authors explored whether the serine protease cathepsin G (CG) might be involved in cortactin degradation. Indeed, the authors could detect CG in neutrophil-stimulated HUVECs. Furthermore, using the supernatant of stimulated neutrophils and incubate the supernatant with stimulated HUVECs led to EC CG detection as well as to reduced cortactin levels. These results indicate that CG might be involved in cortactin degradation. To further explore whether released CG come from azurophilic granule secretion, Rab27 dependent azurophilic granule mobilization and release was blocked with NEI20. This led to a partial reduction in EC cortactin degradation suggesting that other mechanisms might be involved in CG release from neutrophils. Alternatively, other NSPs might be involved as well.

Overall, the ms is well written and the results are quite interesting. Please find below some major and minor issues, which should be addressed by the authors. If the authors are able to address these points, the revised ms could also be submitted as full-length article, in my opinion.

1) You might want to change the title. You did not conclusively show that neutrophil extravasation is dependent on cortactin loss. You only demonstrate that there is a negative correlation btw EC cortactin and neutrophil extravasation. Focus of the ms is on EC cortactin and CG-dependent cortactin degradation.

Authors' response: Thanks. We changed the title to: Neutrophil serine proteases degrade endothelial cortactin and promote extravasation.

2) One major component missing is what signal triggers the release of cathepsin G or other NSP from neutrophils during the recruitment process. This could be a local signal triggered

through interaction of neutrophils with endothelial adhesion molecules (f.e. E-selectin, ICAMs) or chemokines presented on the inflamed endothelium. The authors could address this in the cremaster muscle using respective blocking abs and f.e. TNF as local stimulus and then look for cortactin loss.

Authors' response: Unfortunately, at present we do not have the resources to study the associated mechanisms in detail, and this is beyond the remit of editorial requirements for resubmission as a Report. We thank the Reviewer for their suggested experiments that, hopefully, we can conduct in the future.

3) Along the same line, systemic components in the circulation might also be involved to trigger CG release. Local vs. systemic could be further addressed in flow chambers using stimulated HUVECs and whole blood. Perfusion of the chambers with whole blood for 30 min and subsequent staining of HUVEC cortactin might help to clarify this. Again, you could use blocking antibodies to see what adhesion molecules are involved in triggering cortactin loss.

Authors' response: Thanks for this helpful suggestion. We performed flow chamber assays that confirmed the data from the other models. We also compared perfusion with blood and isolated neutrophils with similar results (Figures 3I-J, and S3B-D), indicating that other cell types in the blood (or soluble components) do not significantly contribute to the cortactin degradation effect. As per above, at present we do not have the resources to study the associated mechanisms in detail and this is beyond the remit of editorial requirements for resubmission as a Report.

Did you check for systemic CG levels in vivo?

Authors' response: No, unfortunately, we do not have the means to do that. However, this has been studied previously, with CatG levels in the peripheral blood of healthy humans being around 500 pg/ml, which increased in coronary artery ectasia to around 680 pg/ml (PMID 26467359). Moreover, cathepsin G has been identified as a critical enzyme involved in neutrophil-inflicted tissue damage during renal ischemia/reperfusion injury (PMID 17322378), and in many other diseases. Thus, CatG indeed plays an important role in neutrophil-mediated inflammatory responses.

4) In my opinion, the static in vitro approach with HUVEC and isolated neutrophils used to confirm the in vivo results from the cremaster muscle might very likely be based on different mechanisms concerning the release of cathepsin G. Here, cell death with passive release of CG could also contribute to the effect. Flow chamber experiments as described in 2) might help to clarify this.

Authors' response: We thank the Reviewer for this comment and indeed we have now conducted a whole new series of experiments using flow chambers. The findings of these studies support our previous data and conclusions (see also point 3).

5) All figures: please add dots from single experiments to all bar graphs.

Authors' response: Done.

March 20, 2026

RE: JCB Manuscript #202410019R-A

Michael Schnoor
Center for Research and Advanced Studies of the National Polytechnic Institute

Dear Dr. Schnoor,

Thank you for submitting your revised manuscript entitled "Neutrophil serine proteases degrade endothelial cortactin and promote extravasation". The manuscript was re-reviewed by the three original referees who all support publication now. We would be happy to publish your paper in JCB pending final revisions necessary to meet our formatting guidelines (see details below). Please also address the final reviewer comment by toning down the conclusions made from figure 4B or provide an appropriate rebuttal.

A. MANUSCRIPT ORGANIZATION AND FORMATTING:

1) Text limits: Character count for Reports is < 20,000, not including spaces. Count includes title page, abstract, introduction, results & discussion, and acknowledgments. Count does not include materials and methods, figure legends, references, tables, or supplemental legends.

2) Figure formatting: Reports may have up to 5 main text figures. Scale bars must be present on all microscopy images, including inset magnifications. Molecular weight or nucleic acid size markers must be included on all gel electrophoresis. Please add scale bars to magnifications in figures 1B/D, 2A/B/C, 5A, S1A and all images in 2C.

Size markers on gels and western blots cannot be the expected sizes of the proteins of interest. In order for readers to accurately assess the size of the proteins shown, the cropped blot images must extend enough to include a region containing at least one of the molecular weight size markers that were run on the gel, which should be labeled. Please revise all blot images as needed.

Also, please avoid pairing red and green for images and graphs to ensure legibility for color-blind readers. If red and green are paired for images, please ensure that the particular red and green hues used in micrographs are distinctive with any of the colorblind types. If not, please modify colors accordingly or provide separate images of the individual channels.

3) Statistical analysis: Error bars on graphic representations of numerical data must be clearly described in the figure legend. The number of independent data points (n) represented in a graph must be indicated in the legend. Please indicate whether 'n' refers to technical or biological replicates (i.e. number of analyzed cells, samples or animals, number of independent experiments). If independent experiments with multiple biological replicates have been performed, we recommend using distribution-reproducibility SuperPlots (please see Lord et al., JCB 2020) to better display the distribution of the entire dataset, and report statistics (such as means, error bars, and P values) that address the reproducibility of the findings.

Statistical methods should be explained in full in the materials and methods. For figures presenting pooled data the statistical measure should be defined in the figure legends. Please also be sure to indicate the statistical tests used in each of your experiments (both in the figure legend itself and in a separate methods section) as well as the parameters of the test (for example, if you ran a t-test, please indicate if it was one- or two-sided, etc.). Also, if you used parametric tests, please indicate if the data distribution was tested for normality (and if so, how). If not, you must state something to the effect that "Data distribution was assumed to be normal but this was not formally tested."

4) Materials and methods: Should be comprehensive and not simply reference a previous publication for details on how an experiment was performed. Please provide full descriptions (at least in brief) in the text for readers who may not have access to referenced manuscripts. The text should not refer to methods "...as previously described." Please also indicate the acquisition and quantification methods for immunoblotting/western blots.

6) For all cell lines, vectors, strains, constructs/cDNAs, etc. - all genetic material: please include database / vendor ID (e.g. Addgene, ATCC, etc.) or if unavailable, please briefly describe their basic genetic features, even if described in other published work or gifted to you by other investigators (and provide references where appropriate). Please be sure to provide the sequences for all of your oligos: primers, si/shRNA, RNAi, gRNAs, etc. in the materials and methods. You must also indicate in

the methods the source, species, and catalog numbers/vendor identifiers (where appropriate) for all of your antibodies, including secondary. If antibodies are not commercial, please add a reference citation or if they have not been published previously, briefly describe how antibodies were generated and validated.

6) Microscope image acquisition: The following information must be provided about the acquisition and processing of images:

- a. Make and model of microscope
- b. Type, magnification, and numerical aperture of the objective lenses
- c. Temperature
- d. Imaging medium
- e. Fluorochromes
- f. Camera make and model
- g. Acquisition software
- h. Any software used for image processing subsequent to data acquisition. Please include details and types of operations involved (e.g., type of deconvolution, 3D reconstitutions, surface or volume rendering, gamma adjustments, etc.).

7) References: There is no limit to the number of references cited in a manuscript. References should be cited parenthetically in the text by author and year of publication. Abbreviate the names of journals according to PubMed.

8) Supplemental materials: Reports may have up to 3 supplemental figures and 10 videos. Please also note that tables, like figures, should be provided as individual, editable files. A summary of all supplemental material should appear at the end of the Materials and methods section. Please include one brief sentence per item.

9) Video legends: Should describe what is being shown, the cell type or tissue being viewed (including relevant cell treatments, concentration and duration, or transfection), the imaging method (e.g., time-lapse epifluorescence microscopy), what each color represents, how often frames were collected, the frames/second display rate, and the number of any figure that has related video stills or images.

10) eTOC summary: A ~40-50 word summary that describes the context and significance of the findings for a general readership should be included on the title page. The statement should be written in the present tense and refer to the work in the third person. It should begin with "First author name(s) et al..." to match our preferred style.

11) Conflict of interest statement: JCB requires inclusion of a statement in the acknowledgements regarding competing financial interests. If no competing financial interests exist, please include the following statement: "The authors declare no competing financial interests." If competing interests are declared, please follow your statement of these competing interests with the following statement: "The authors declare no further competing financial interests."

12) A separate author contribution section is required following the Acknowledgments in all research manuscripts. All authors should be mentioned and designated by their first and middle initials and full surnames. We encourage use of the CRediT nomenclature (<https://casrai.org/credit/>).

13) ORCID IDs: ORCID IDs are unique identifiers allowing researchers to create a record of their various scholarly contributions in a single place. Please note that ORCID IDs are required for all authors. At resubmission of your final files, please be sure to provide your ORCID ID and those of all co-authors.

14) JCB requires authors to submit Source Data used to generate figures containing gels and Western blots with all revised manuscripts. This Source Data consists of fully uncropped and unprocessed images for each gel/blot displayed in the main and supplemental figures. For assays performed using capillary electrophoresis and/or immunoassay-based detection, authors should instead provide the electropherogram graph(s) for each experiment, plotting fluorescence/chemiluminescence intensity vs. molecular weight/size. Since your paper includes cropped gel and/or blot images, please be sure to provide one Source Data file for each figure gels, blots, and/or capillary electrophoresis assays along with your revised manuscript files. File names for Source Data figures should be alphanumeric without any spaces or special characters (i.e., SourceDataF#, where F# refers to the associated main figure number or SourceDataFS# for those associated with Supplementary figures). For traditional gels and blots, the lanes of the gels/blots should be labeled as they are in the associated figure, the place where cropping was applied should be marked (with a box), and molecular weight/size standards should be labeled wherever possible. For capillary electrophoresis assays, each trace in the graph should be color-coded and labeled to indicate which protein, gene, or sample is being measured (please try to avoid red/green combinations to accommodate our color-blind readers).

Source Data files will be directly linked to specific figures in the published article. Source Data Figures should be provided as individual PDF files (one file per figure). Authors should endeavor to retain a minimum resolution of 300 dpi or pixels per inch. Please review our instructions for export from Photoshop, Illustrator, and PowerPoint here: <https://rupress.org/jcb/pages/submission-guidelines#revised>.

15) Journal of Cell Biology now requires a data availability statement for all research article submissions. These statements will be published in the article directly above the Acknowledgments. The statement should address all data underlying the research

presented in the manuscript. Please visit the JCB instructions for authors for guidelines and examples of statements at (<https://rupress.org/jcb/pages/editorial-policies#data-availability-statement>).

B. FINAL FILES:

The license to publish form must be signed before your manuscript can be sent to production. A link to the license to publish form will be sent to the corresponding author only. Please take a moment to check your funder requirements before choosing the appropriate license.

Thank you for your attention to these final processing requirements. Please revise and format the manuscript and upload materials within 7 days. If you need an extension for whatever reason, please let us know and we can work with you to determine a suitable revision period.

Thank you for this interesting contribution, we look forward to publishing your paper in Journal of Cell Biology.

Sincerely,

Anna Huttenlocher, MD, PhD
Monitoring Editor
Journal of Cell Biology

Dan Simon, PhD
Scientific Editor
Journal of Cell Biology

Reviewer #1 (Comments to the Authors (Required)):

all comments have been addressed and the paper has been improved.

Reviewer #2 (Comments to the Authors (Required)):

The authors now show flow cytometry results, demonstrating that cathepsin G is detectable only in permeabilized ECs. The resulting figures 3I and J are convincing and are probably sufficient to support the main conclusion. However, figure 4B is not. Airyscan does not have sufficient resolution.

Reviewer #3 (Comments to the Authors (Required)):

The authors have adequately addressed my concerns. The new experiments helped to strengthen the data. Therefore, I do not have any additional comments.